



# A Lagrangian study of the contribution of the Canary coastal upwelling to the open North Atlantic nitrogen budget

Derara Hailegeorgis[1], Zouhair Lachkar[1], Christoph Rieper[2,3], and Nicolas Gruber[2]

[1]Center for Prototype Climate Modeling, New York University Abu Dhabi, Abu Dhabi, UAE
[2]Environmental Physics, Institute of Biogeochemistry and Pollutant Dynamics, ETH Zurich, Universitätstrasse 16, 8092 Zurich, Switzerland
[3]Experimental Oceanography, Institute of Oceanography, University of Hamburg, Bundesstrasse 53, 20146 Hamburg, Germany

**Correspondence:** Zouhair Lachkar (zouhair.lachkar@nyu.edu)

**Abstract.**

The Canary Current System (CanCS) is a major Eastern Boundary Upwelling System (EBUS), known for its high nearshore productivity and for sustaining large fisheries. Only a part of the inorganic nutrients that upwell along Northwest Africa are being used to fuel the high nearshore productivity. The remainder together with some of the newly formed organic nutrients are

exported offshore into the adjacent oligotrophic subtropical gyre of the North Atlantic. Yet, the offshore reach of these nutrients and their importance for the biogeochemistry of the open North Atlantic is not yet fully quantified. Here, we determine the lateral transport of both organic and inorganic nitrogen from the Canary upwelling and investigate the timescales, reach, and structure of offshore transport using a Lagrangian modelling approach. To this end, we track all water parcels entering the coastal ocean and upwelling along the Northwest African coast between 14°N and 35°N, as simulated by an eddy-resolving

configuration of the Regional Ocean Modeling System (ROMS). Our model analysis suggests that the vast majority of the upwelled waters originate from offshore and below the euphotic zone (70 m depth), and once upwelled remain in the top 100m. The offshore transport is intense, yet it varies greatly along the coast. The central CanCS (21°N-28°N) transports the largest amount of water offshore, thanks to a larger upwelling volume and a faster offshore transport. In contrast, the southern CanCS (14°N-21°N) exports more nitrogen from the nearshore, primarily because of the higher nitrogen-content of its upwelling

waters. Beyond 200 km, this nitrogen offshore transport declines rapidly because the shallow depth of most water parcels supports high organic matter formation and subsequent export of the organic nitrogen to depth. The horizontal pattern of offshore transport is characterized by latitudinally alternating offshore-onshore corridors indicating a strong contribution of mesoscale eddies and filaments to the mean transport. Around 1/3 of the total offshore transport of water occurs around major capes along the CanCS. The persistent filaments associated with these capes are responsible for an up to four-fold enhancement

of the offshore transport of water and nitrogen in the first 400km. Much of this water and nitrogen stems from upwelling at quite some distance from the capes, confirming the capes' role in collecting water from along the coast. North of Cape Blanc and within the first 500 km from the coast, water recirculation is a dominant feature of offshore transport. This process, likely associated with mesoscale eddies, tends to reduce the efficiency of offshore transport. This process is less important in the southern CanCS along the Mauritanian coast. The Canary upwelling is modelled to supply around 44 mmol N m$^{-2}$yr$^{-1}$ and





7 mmol N m$^{-2}$yr$^{-1}$ to the North Atlantic Tropical Gyral (NATR) and the North Atlantic Subtropical Gyral East (NASE) Longhurst provinces, respectively. In the NATR, this represents nearly half (45±15%) of the estimated total new production, while in the NASE, this fraction is small (3.5±1.5%). Our results highlight the importance of the CanCS upwelling as a key source of nutrient to the open North Atlantic and stress the need for improving the representation of EBUS in global coarse
resolution models.

## 1   Introduction

Over the last three decades, several studies have highlighted the importance of the coastal ocean in the global carbon cycle (e.g., Walsh 1991; Wollast, 1998). For instance, it has been estimated that continental margins contribute nearly half of the globally integrated oceanic primary production, although they only occupy about 10% of the world ocean surface (Walsh,
1991; Smith and Hollibaugh, 1993; Muller-Karger et al., 2005; Liu et al., 2010; Jahnke, 1990). The coastal ocean is not only highly productive, but is also a major source of nutrients and organic matter to the open ocean (Liu et al., 2000, 2010; Lovecchio et al., 2017; Frischknecht et al, 2018). For example, Wollast (1998) and Ducklow and McCallister (2004) estimated that about half of the total production in the coastal ocean is exported laterally, leading to substantial modifications of the biogeochemical cycles there. Globally, the magnitude of this lateral export of organic and inorganic matter from the coastal ocean into the
open ocean is ill constrained and highly debated. Additionally, little is known about the exact fate and impact of this laterally exported matter in the open ocean. It is generally assumed that most of the organic matter eventually gets remineralized back to its inorganic constituents. In extreme cases, the associated increase in heterotrophic activity may actually exceed the amount of autotrophic production, making such systems net heterotrophic. Potential places for this to occur are the subtropical gyres where strong nutrient limitations leads to low levels of productivity (Del Giorgio et al., 1997; Duarte and Agusti, 1998). The
coastal ocean represents also a potentially important, although not well understood conduit in the global carbon and nutrient cycles. Especially low-latitude upwelling margins could play a crucial role in the return pathway of carbon and nutrients from the deep ocean back to the low-latitude surface ocean, which is needed in order to compensate for the continuous loss of these elements from the low latitude ocean through sinking into the deep ocean (Gruber and Sarmiento, 2002). Yet, despite decades of research, our ability to constrain this conduit in a quantitative manner has remained very limited (cf. Holzer and Primeau,
25   2008).

The Eastern Boundary Upwelling Systems (EBUS) play a particularly important role for the global exchange of organic and inorganic matter between the coastal and open oceans. These systems are among the most productive ecosystems in the world and sustain a high fraction of global export production and global fish catch (Pauly and Christensen, 1995; Carr, 2001; Hansell, 2002; Chavez and Messie, 2009). Their productivity is fuelled by nutrient-rich water that upwells at the coast due to alongshore
equatorward winds. Although the four major EBUS in the world make up a small part of the global ocean (0.1% - 0.3% in most studies), they account for 30% of the world's fish catch (Durand et al. 1998; Carr and Kearns, 2003; Rykaczewski and Checkley, 2008; Arístegui et al. 2009). The lateral transport of inorganic and organic matter is potentially particularly relevant for EBUS, as they export this material into the adjacent subtropical gyres. These gyres are likely particularly receptive as they



have low nutrient conditions (oligotrophic) and have been shown to be net heterotrophic based on observational data (Duarte and Agustí, 1998; Del Giorgio and Duarte, 2002). Yet, the question of whether and how far into the open ocean the additional input of organic and inorganic matter can increase in situ respiration and new production (NP) remains a subject of a long and unresolved debate (Smith and Hollibaugh, 1993; Duarte and Agustí, 1998; Liu et al., 2010; Lovecchio et al., 2017).

Located along the northwestern African coast, the Canary Current System (CanCS) constitutes the EBUS of the North Atlantic subtropical gyre (Pelegri et al., 2005a; Pelegri et al., 2006; Barton, 1989). The CanCS is composed of the Canary Current (CC) and the Canary Upwelling Current (CUC). The CC represents the eastern boundary current of the North Atlantic Subtropical Gyre. It flows parallel to the Moroccan coast and merges with the westward North Equatorial Current (NEC) around Cape Blanc (21°N) (Barton, 1987; Hernandez-Guerra et al., 2005). The CUC is a nearshore surface jet associated

with the upwelling flowing equatorward along the northwest African coast (Pelegri et al., 2006). The Cape Verde frontal zone, between Cape Blanc and the Cape Verde archipelago, is dominated by a permanent cyclonic circulation with a poleward boundary current, the Mauritanian current (MC), that extends at depth beyond Cape Blanc as a slope undercurrent typical of eastern boundary upwelling systems (Barton et al., 1989; Arístegui et al., 2009). Upwelling is permanent along most of the Moroccan coast (21-35°N), albeit with weaker intensity and stronger seasonality north of 26°N (Cropper et al., 2014). South of

Cape Blanc, upwelling is present essentially in late fall and winter (October-March). Upwelled waters have different nutrient contents depending on their respective sources. North of Cape Blanc, the relatively nutrient-impoverished North Atlantic Central Waters (NACW) feed most of the upwelling. In contrast, waters upwelling south of Cape Blanc have a higher nutrient content as they are fed by the nutrient-richer South Atlantic Central Waters (SACW).

     Intense mesoscale structures, including eddies and filaments develop in different parts of the CanCS and contribute to the

offshore transport of the coastal waters (Lovecchio et al., 2018). Àlvarez-Salgado et al. (2007) estimate that persistent coastal filaments at the West African coast and Iberia export carbon at a rate 2.5 - 4.5 times higher than Ekman transport. Pelegri et al. (2005a) found that in the Canary Basin coastal filaments and cyclonic eddies cause localized offshore export of nutrients and organic carbon. The role of upwelling filaments in the shelf to open ocean transport of organic matter depends also on eddy-filament interactions, which frequently occur on ocean margins (e.g. Barton et al., 1998; Brink and Cowles, 1991). Oceanic

eddies may entrain filament waters with higher content of organic matter, enhancing the shelf-ocean exchange (Aristegui et al., 1997). Filaments may return to the continental shelf part of the water upwelled and expelled from the coast by means of eddy-associated circulation. This recirculation decreases the impact of filaments on the offshore transport of organic matter (e.g. Basterretxea and Aristegui, 2000).

     The quantification of the magnitude and type of lateral export is challenging due to the complex biogeochemical and physical

dynamics of the offshore transport. Much of it is being driven by mesoscale processes, whose appropriate in-situ sampling goes beyond the abilities of current observing systems, thus requiring the use of high-resolution coupled physical/biogeochemical models. Yet, the number of high-resolution model-based studies that addressed and quantified the coastal-open ocean exchange in the CanCS remain limited (e.g., Fischer and Karakas, 2009; Lachkar and Gruber, 2011; Pastor et al., 2013; Auger et al., 2016; Lovecchio et al., 2017; Lovecchio et al., 2018). Such studies have stressed the importance of eddies and coastal filaments in

the offshore transport of coastal upwelling. Lovecchio et al. (2017), for example, examined the export of organic carbon from





the CanCS using a coupled physical-biogeochemical ocean model with a telescopic grid that covers the whole Atlantic ocean while maintaining a high resolution along the coast of northwest Africa. They demonstrated that about a third of the organic carbon produced along the northwest African coast is transported offshore, and some of it well beyond 1500 km from the coast, contributing substantially to the net community production there. In a follow-up study, Lovecchio et al. (2018) showed that

much of this transport is driven first by filaments (in the first 100 km from the coast), and then later taken over by westward propagating mesoscale eddies. However, these authors focused on the transport of organic carbon only, and therefore did not estimate the contribution of the Canary upwelling to the offshore export of nutrients into the open ocean and the implications this might have for the biogeochemistry of the North Atlantic Ocean. Furthermore, the Eulerian approach used by Lovecchio et al. (2017, 2018) does not allow for the identification of the contribution of the upwelled waters. It is also not well suited for

the identification and quantification of the pathways of the coastal-open ocean exchange.

Here, we aim to close this gap, and investigate the contribution of the upwelling waters to the nitrogen budget of the open North Atlantic, thereby considering the transport of both organic and inorganic forms of nitrogen. To this end, we use a Lagrangian approach to quantify the reach, the spatial structure and the timescales of the offshore transport of upwelled waters by tracking all open ocean waters that upwell along the coastal region of the CanCS. By not only tracking waters through the

appropriate seeding of Lagrangian particles, but also tabulating the biogeochemical transformations along the pathways, we can also establish Lagrangian budgets. This permits deep insights into the working of the CanCS system and its connection to the open ocean.

Previous studies have used the Lagrangian approach to study different aspects of the CanCS. For instance, Brochier et al. (2011) conducted a ROMS-based Lagrangian experiment to study the transport of ichthyoplankton (fish eggs and larvae)

due to filaments between the West African coast and the Canary Islands. Mason et al. (2012) used a Lagrangian approach to characterize the source waters of upwelling in the CanCS between 31°N and 35°N. Yet, these studies were limited to specific regions of the CanCS and did only partially sample coastal upwelling there. Here we substantially expand on these previous efforts by sampling and tracking all open ocean waters that upwell along the West African coast between 14°N and 35°N, and quantify the offshore export of water, nutrients and organic matter. We investigate the kinetics and the structure of this offshore

transport and explore the role of water recirculation and capes in enhancing both coastal upwelling and offshore export. Finally, we examine the contribution of the CanCS upwelling to the Open North Atlantic nitrogen budget.

## 2   Methods

### 2.1   Model setup

#### 2.1.1   Models and configuration

We use a CanCS configuration of the Regional Ocean Modeling System (ROMS)-AGRIF (http://www.croco-ocean.org/) similar to that used by Lachkar et al. (2016). ROMS solves the primitive equations and has a free-surface and a terrain-following vertical coordinates (Shchepetkin and McWilliams, 2005). We use a rotated-split third-order upstream biased operator for



the advection of momentum and material properties (Marchesiello et al., 2009). The non-local K-profile parameterization (KPP) scheme is used to represent the subgrid vertical mixing (Large et al., 1994). The biogeochemical model is a nutrient-phytoplankton-zooplankton-detritus (NPZD) model based on nitrogen (Gruber et al., 2006). It uses a system of ordinary differential equations representing the time-evolution of the following state variables: nitrate ($NO_3^-$), ammonium ($NH_4^+$), phyto-

plankton, zooplankton, two pools of detritus and a dynamic chlorophyll-to-carbon ratio. The two classes of detritus represent, respectively, fast-sinking large organic matter particles and slow-sinking small particles. The small particles can coagulate with phytoplankton to form large detritus.

The model domain covers the region from 10°N to 42°N in latitude and from 30°W to 6°W in longitude with a grid resolution of 1/20°. This corresponds to a mesh size of about 5 km, which is sufficient for fully resolving mesoscale processes. The vertical

grid consists of 32 layers with enhanced resolution near the surface. The bathymetry is derived from the ETOPO2 file provided by the National Geophysical Data Center (Smith and Sandwell, 1997). We use a monthly climatological forcing based on the Comprehensive Ocean, Atmosphere Data Set (COADS) (da Silva et al., 1994) for surface heat and freshwater fluxes. Surface temperature and salinity are restored to COADS observations using kinematic heat and freshwater flux corrections following Barnier et al. (1995). Wind stress is derived from the QuikSCAT-based Scatterometer Climatology of Ocean Winds (Risien

and Chelton, 2008). The initial and lateral boundary conditions for temperature, salinity and nitrate are derived from the World Ocean Atlas (WOA) 2009. Other ecological tracers are initialized uniformly to arbitrary low values. Currents at the boundaries are derived from temperature and salinity data using geostrophy together with Ekman transport in the upper 40 m. The model starts from rest and is spun up for 12 years. We use the last 3 years of the simulation (i.e., Year 10 to Year 12) for analysis. Model outputs are stored at daily frequency.

### 2.1.2 Model evaluation

The model successfully simulates the Canary current flowing parallel to the Moroccan shore and its detachment from the coast and merger with the westward NEC around the latitude of Cape Blanc (Fig. 1). However, the strength of both the northward coastal Mauritanian current in the south (16-20°N) and the eastward Azores current in the north (34°N) is underestimated in the model in comparison to the drifter data. The model simulates successfully the surface Eddy kinetic energy (EKE, Fig. 1)

in the central and southern subregions as it reproduces quite accurately the high-EKE values observed downstream the Canary islands and the Cape Verde Archipelago and around the Cape Verde Peninsula in Senegal (Fig. 1). However, the model tends to underestimate EKE associated with the Azores current in the northern part of the domain. This might be due to the too weak Azores current in our model as well as the lack of high-frequency variability in the employed (climatological) forcing.

Despite a large-scale cold bias of around 1°C throughout most of the domain, the model captures the main patterns of the

observed sea-surface temperature (SST) from the Advanced Very High Resolution Radiometer (AVHRR) satellite data (Fig. 2). In particular, the model reproduces the observed offshore gradient in SST, as well as the offshore extent of the cold upwelling region. The model simulates the spatial pattern of the observed chlorophyll-a (SeaWiFS, Fig 2) reasonably well over much of the domain, with high-concentrations in the upwelling region and lower concentrations in the open oligotrophic ocean (Fig. 2). The north-south chlorophyll gradient is also well reproduced with the southern part of the Canary system being more





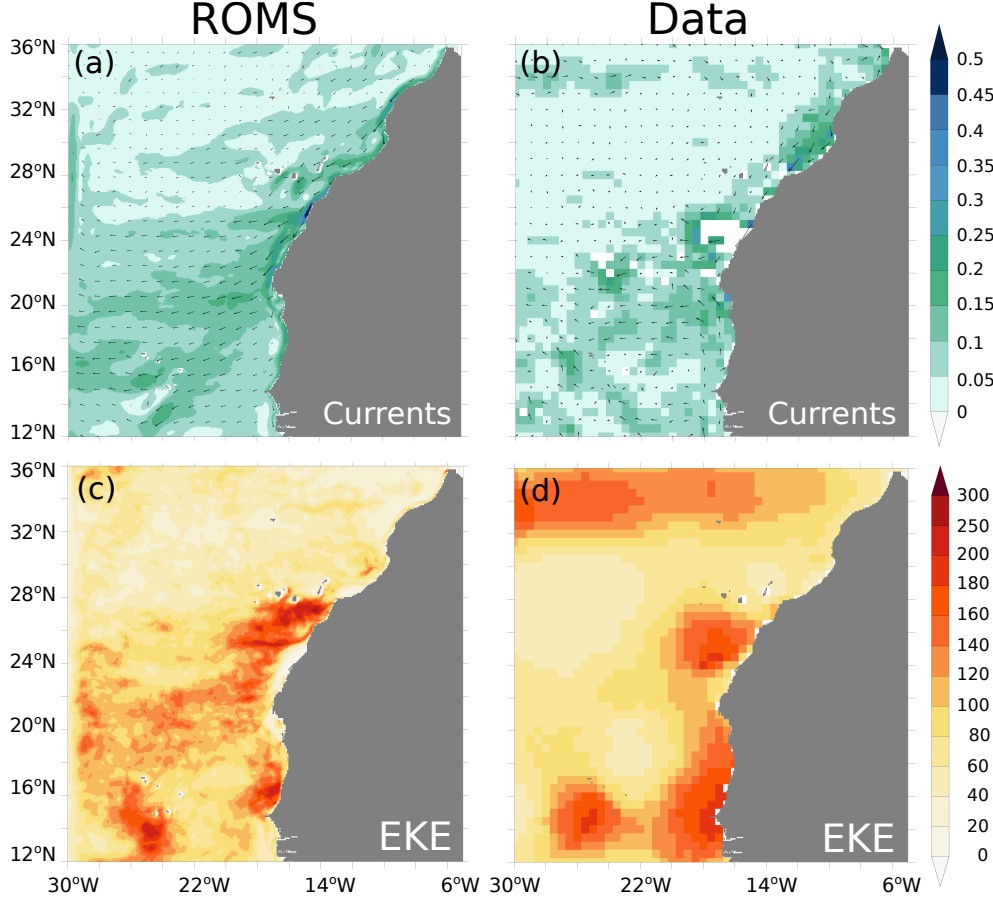

**Figure 1.** Surface (a-b) currents (in m s$^{-1}$) and (c-d) eddy kinetic energy (in cm$^2$ s$^{-2}$) as simulated in the model (left) and from surface drifter climatology of Lumpkin and Johnson (2013) (right). (a-b) Arrows indicate the direction of the current and the color shading shows the current magnitude.

productive than the northern part. However, the model underestimates the observed cholorophyll-a in the nearshore regions, particularly in the southern part of the domain.

A more quantitative evaluation of the model skill can be achieved using both satellite data and in situ observations from the World Ocean Database 2018 (WOD 2018). We binned the latter onto a 0.5° monthly climatology without any spatial

5  interpolation. This permits us to compare the model and observations only in those grid points where observations are available. The results of this evaluation are summarized graphically using Taylor diagrams (Taylor, 2001) quantifying the similarity between the observations and model in terms of their correlation, the amplitude of their standard deviations and their centered root mean square difference (rmsd) (Fig. 3). We find that the simulated and observed mean SST have similar standard deviations and correlate very strongly with correlation coefficients above 0.95 for both nearshore and offshore regions (Fig. 3a). Similarly,

10  the quantitative comparison of the simulated and observed sea surface salinity (SSS) shows similar standard deviations and a




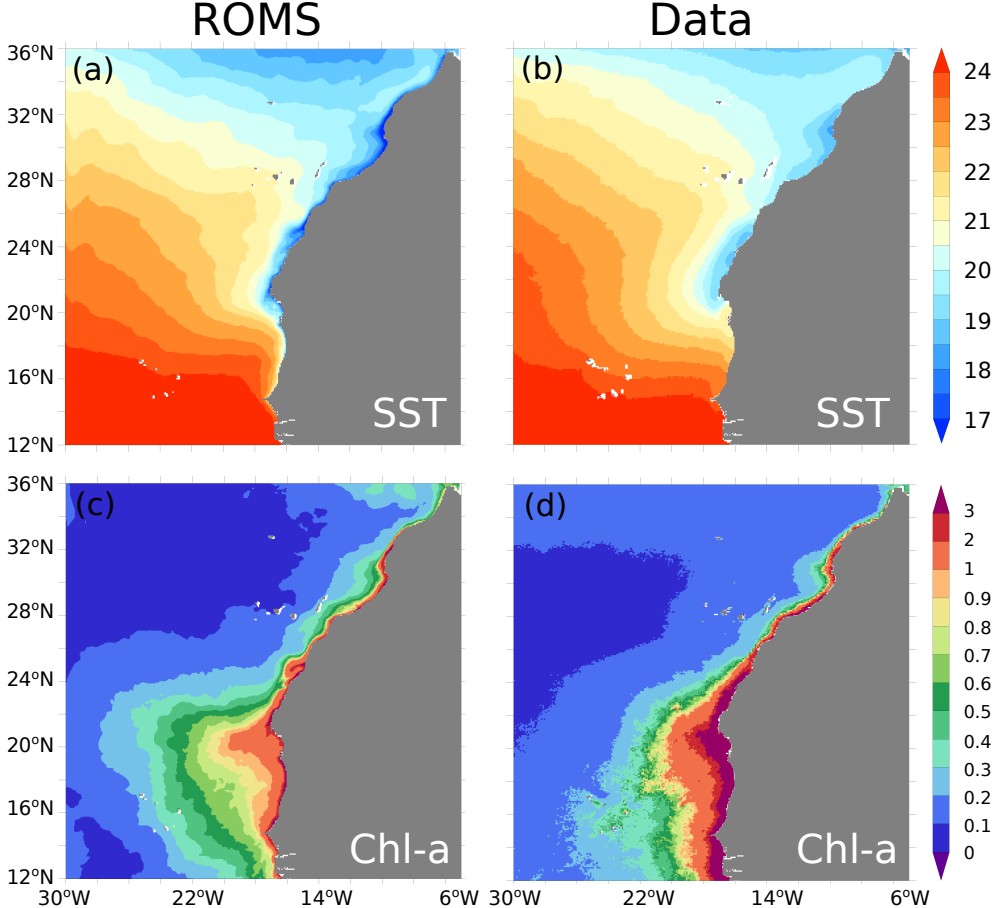

**Figure 2.** Sea surface (a-b) temperature (in °C ) and (c-d) chlorophyll-a concentrations (in mg m$^{-3}$) as simulated in ROMS (left) and from AVHRR and SeaWiFS data (right). The AVHRR and the SeaWiFS climatologies are computed over the periods from 1981 to 2016 and from 1997 to 2009, respectively.

small rmsd, as well as very high correlations (r > 0.9) in the nearshore and throughout the domain. Mean surface nitrate in the model has a comparable variance but a weaker correlation (r ≈ 0.7) with observations. Simulated mean surface chlorophyll-a is moderately (0.6 < r < 0.8) correlated with observations and displays a substantially weaker variance than the satellite data.

The seasonal anomaly of SST in the model agrees well with observations at all distances from the coast with both the corre-
5   lation coefficient and normalized standard deviation being over 0.9 (Figure 3b). SSS seasonal anomalies in the model correlate less strongly with observations with both correlation coefficients and normalized standard deviations having values below 0.7. Simulated surface nitrate seasonal anomalies show very similar variance in comparison to observations but correlate only weakly with data (0.3 < r < 0.5). The simulated surface chlorophyll-a seasonal anomalies have a similarly weak correlation with the observations in addition to a lower normalized standard deviation (below 0.5).



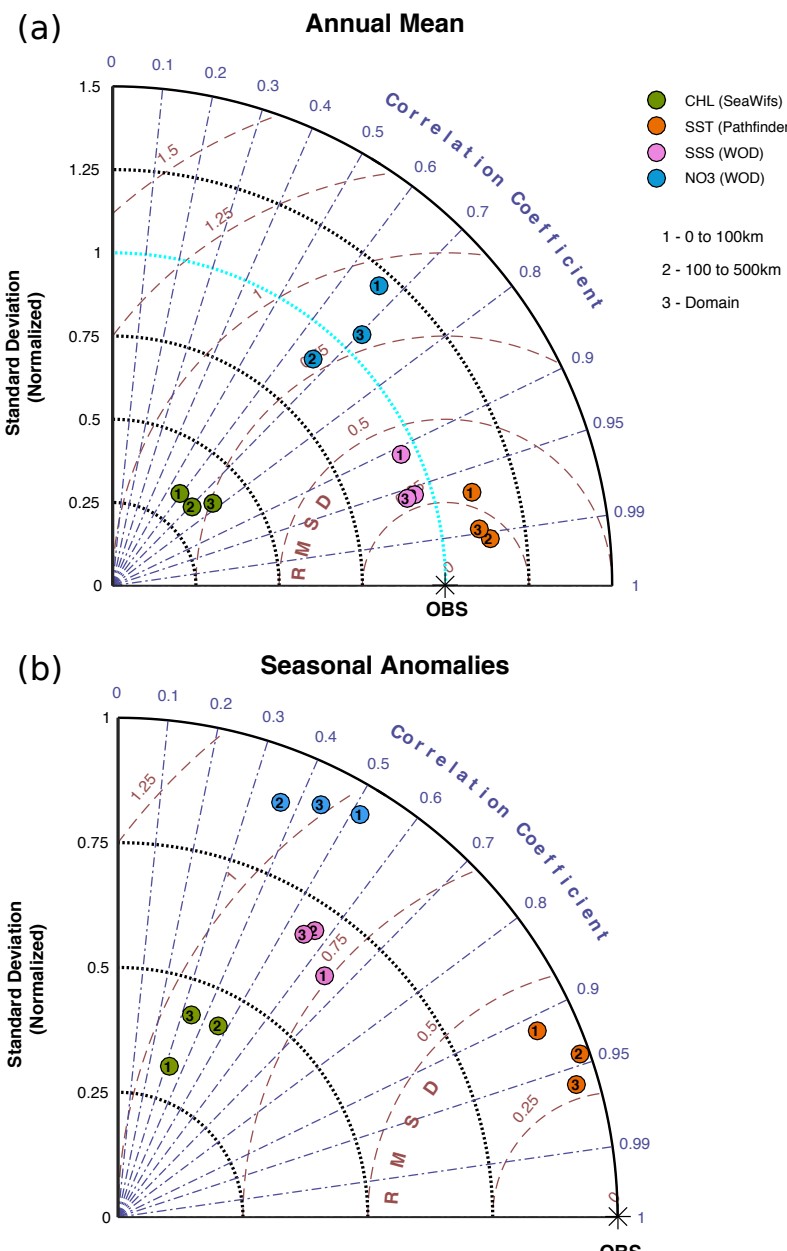

**Figure 3.** Taylor diagrams describing statistical comparisons of modeled and observed (a) annual mean and (b) seasonal anomaly estimates of sea surface temperature (orange), salinity (pink), $NO_3^-$ (blue) and chlorophyll-a (green) in the 0-100 km nearshore region (data points labeled "1"), the 100-500 km offshore region (data points labeled "2") and across the entire model domain (data points labeled "3"). The reference point of the Taylor diagram corresponds to SeaWiFS observations for chlorophyll, AVHRR data for sea surface temperature, world ocean database (WOD) 2013 for surface salinity and nitrate.





We finally evaluate the model ability to reproduce the three-dimensional distributions of temperature, salinity and nitrate in the upper ocean (Figure 2 in Supplementary Information (SI)). At all depth ranges (top 100 m, top 400 m and top 1000 m), mean temperature, salinity and nitrate are all reproduced quite well with correlations above 0.9 and normalized standard deviations near 1. The seasonal anomalies (shown only for the depth range of 0 - 400 m), are less accurately captured by the model.

Temperature anomaly in the top 400 m is captured relatively well with a correlation close to 0.8 and a standard deviation comparable to observations. Salinity and nitrate anomalies show lower correlations (0.3-0.4), diverge further in rmsd (close to 1) and have a narrower distribution with a standard deviation of around 0.6.

In summary, despite some local biases, the model generally shows good skill in reproducing the large-scale features of the circulation and the productivity of the Canary current region. More importantly, it reproduces well the strength and structure

of the Canary coastal upwelling. Overall, this CanCS-only ROMS setup has similar strengths and weaknesses as the telescopic grid setup employed by Lovecchio et al. (2017, 2018). We will discuss the potential impact of the model limitations on our results in the discussion section.

## 2.2 Lagrangian experiment

The Lagrangian particle tracking experiment is performed offline with ARIANE (Blanke and Raynaud, 1997; http://stockage.univ-

brest.fr/ grima/Ariane/) using ROMS daily output. ARIANE tracks water particles based on the velocity output of the model. In this experiment, ARIANE runs based on the model output of zonal and meridional velocities (ARIANE internally computes vertical velocities from the continuity equation). ARIANE analytically computes streamlines of particle trajectories across grid-walls by assuming a steady-state flow. The data points for successive days are used as piecewise steady flow where the experiment is assumed to be static for each day and not interpolated between different days. The velocity at a given point inside

the regular cells of the model is computed by linearly interpolating the velocity at opposite faces of a cell.

With our Lagrangian experiment we aim to study trajectories of open ocean water masses that enter the coastal region and upwell between 14°N and 35°N. This region extends in north-south direction a total of 3185 km and covers most of the West African coast component of the Canary Current Coastal Province (CNRY) of Longhurst et al. (2007) (Fig. 4a, Fig 3 in SI). The control box is bounded by the particle release strip (both entry and exit) to the west, the coast to the east and particle exit (but

not entry) sections at the northern and southern ends of the upwelling coastal region (Figure 4a). Since particles are released only along the western segment of the control box, only open ocean particles that enter the coastal region are tracked and considered candidates for upwelling. The 200 m isobath along the coast indicating the limit of the continental shelf is identified and particles are released up to 50 km further west from it (Fig. 4b). This ensures sampling coastal upwelling occurring at the shelf break. This results in a coastal strip with a width that varies from below 60 km near 27°N where the the continental shelf

is narrowest to over 160 km near 25°N where the shelf is widest. Particles are released on a daily basis during one year and their trajectories are tracked during a two-year period. We limit the maximum transport a particle is associated with to 0.01 Sv, which limits the maximum volume assigned to one particle to 0.864 km$^3$. In total 9,888,387 particles are released over the first year (the period of particle release).



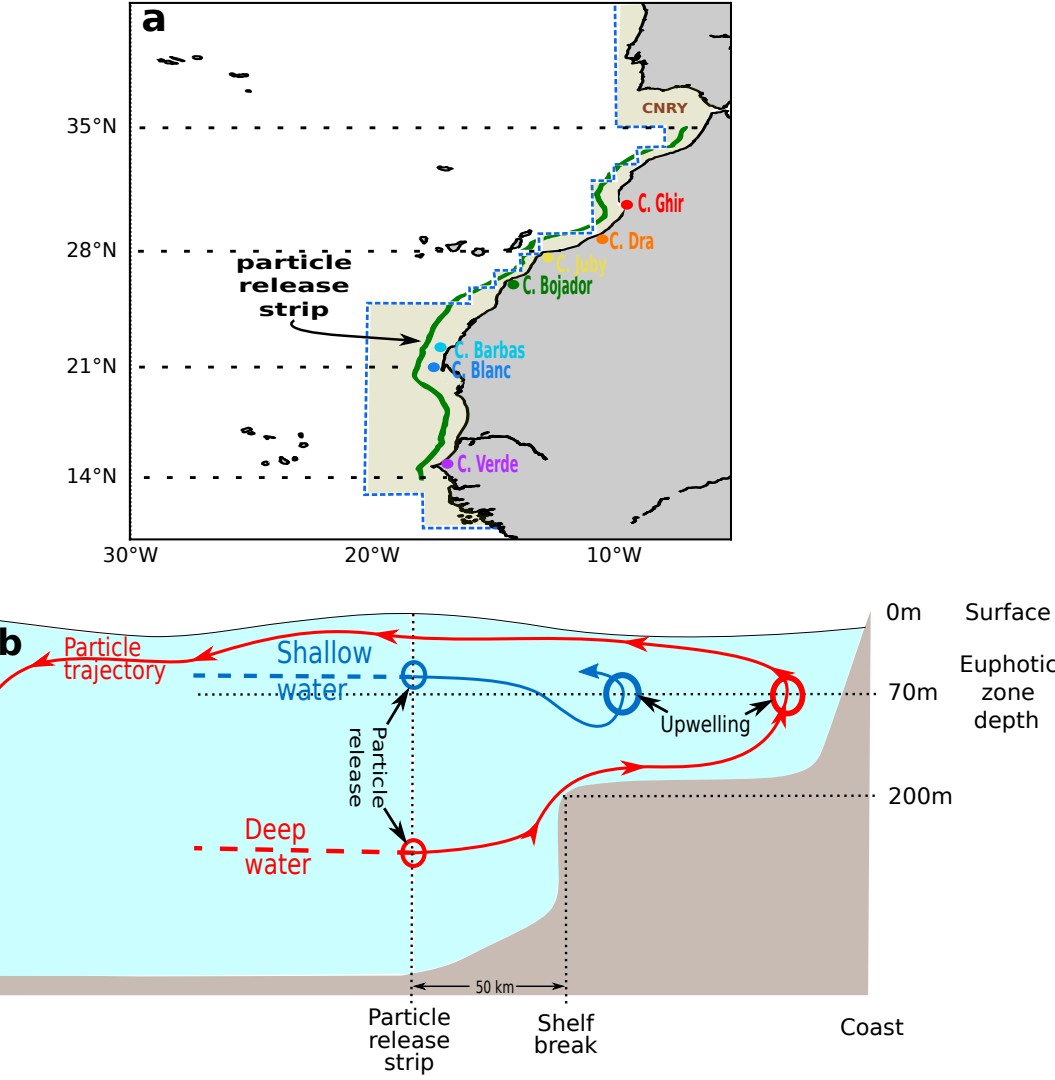

**Figure 4.** (a) The model domain stretches from 30°W to 6°W and from 10°N to 41°N. The solid green line indicates the limit of the coastal strip where particles are released at each vertical layer proportionally with the onshore water volume transport. The black dotted lines partition the domain into the three latitudinal subregions: the southern CanCS (14°N - 21°N), the central CanCS (21°N - 28°N) and the northern CanCS (28°N - 35°N). The yellow shaded area corresponds to the portion of the Longhurst Canary Coastal Province (CNRY) that lies within the model domain. The location of the seven capes studied in Section 4.1 is shown here in different colors. (b) A schematic of the Lagrangian sampling of upwelling particles. Particles are released when open ocean water enters the coastal strip. Based on their depth when released, we term particles to be of shallow ($<$ 70 m) or deep ($>$ 70 m) source. Since their release into the coastal region, particles are marked as upwelling the first time they cross the 70 m depth and their trajectory tracked thereafter.





Along each particle's trajectory, its latitude, longitude and depth are recorded along with biogeochemical tracers and the water volume tagged with each particle at its release that is assumed to follow the particle throughout its trajectory. Assuming the ocean to be an incompressible fluid and its velocity field to be non-divergent, the volume transport is conserved . This allows ARIANE to compute particle trajectories and volume transports both forward and backward in time. We quantify the

amount of nutrients carried by each particle to a given location as the product of its associated volume and the concentration of the tracer associated with the particle when it reaches that location. Tracked particles that enter the coastal region and cross the 70 m depth to move up to shallower depths are identified as upwelling particles. Since only particles that enter the coastal area and upwell there are followed, our experiment disregards wind stress curl-driven upwelling that occurs in the open ocean and can locally be important, particularly in the southern subregion (Lovecchio et al., 2017). From all tracked particles, 352,873

(3.57%) upwell. These are the particles that we follow thereafter and form the basis for our analyses (see additional details of the Lagrangian experiment in SI).

Previous studies have shown subregional differences within the CanCS in circulation patterns, mesoscale activity, seasonality of upwelling, biology and sub-surface nutrient concentration (Arístegui et al., 2009; Pelegri and Benazzouz, 2015; Lovecchio et al., 2017). We thus separately characterize the trajectories of waters that upwell in three latitudinal subregions of the upwelling

domain. These subregions cover southern (14°N - 21°N), central (21°N - 28°N) and northern (28°N - 35°N) parts of the CanCS (Figure 4a).

## 3   Characteristics of upwelling

### 3.1   Upwelling of water

Our Lagrangian analysis reveals an annual upwelling of offshore-derived waters of nearly 80,000 km$^3$ (see Table 1). This is

about 25 km$^3$ of water per year per kilometer of coastline, or 0.8 m$^3$ s$^{-1}$ per meter of coastline. The strongest upwelling occurs in the central CanCS, with a particularly strong peak around Cape Bojador (27°N) (Fig. 5a). This subregion alone is responsible for more than half of the total upwelling (40,705 km$^3$ yr$^{-1}$). The northern subregion contributes about 25,000 km$^3$ per year to the upwelling, while the southern subregion has the smallest upwelling (13,000 km$^3$ yr$^{-1}$).

|  | Particles | Water (km$^3$) | N (Gmol) | Median dist (km) | To 400km (%) | To 1000km (%) |
|---|---|---|---|---|---|---|
| Southern | 51,275 | 13,178.9 | 337.7 | 64.1 | 95.4 | 77.8 |
| Central | 162,304 | 40,704.8 | 210.3 | 37.1 | 92.5 | 69.3 |
| Northern | 139,294 | 25,038.0 | 86.8 | 41.1 | 71.7 | 33.1 |
| Whole CanCS | 352,873 | 78,921.7 | 634.8 | 42.9 | 86.3 | 59.2 |

**Table 1.** The annual number of upwelling particles and their associated water and nitrogen, the median distance of upwelling to the coast and the net transport of water to 400 km and 1000 km from coast (as percent of upwelling volume) in the three subregions as well as the entire CanCS region in our experiment.





**Figure 5.** (a) Zonally integrated annual upwelling water volume. (b,c,d) Annual upwelling water volume with distance from the coast in the (b) southern, (c) central and (d) northern subregions. Light and dark blue segments represent contribution by deep and shallow sources of water, respectively.



Most of the upwelling stems from deeper waters, i.e., waters that enter our analysis region below 70 m. 85% of the tracked particles and 88% of the associated water volume follow that path (termed 'deep source'). The remaining upwelling stems from waters that enter the region within the euphotic zone and then are transported below 70 m before upwelling (here termed 'shallow source') (Figure 4b). This fractional distribution varies little between subregions.

But the offshore distribution of the upwelling differs strongly between the three CanCS subregions (Fig. 5). While the maximum of the upwelling in the Southern subregion occurs offshore at round 70 km distance from the shore with a weak onshore-offshore contrast, the central subregion has a clear maximum in upwelling in the first 20 km from the shore. The amount of upwelling then declines sharply with increasing offshore distance. The northern subregion has a first upwelling volume maximum right at the coast, followed by a second one at around 50 km from the shore. Beyond 80 km, there is hardly

any upwelling. These differences are well reflected in the median upwelling distances being 64.1km, 37.1km and 41.1 km in the southern, central and northern subregions, respectively (Table 1).

Most of this variability has to do with the variability of offshore Ekman transport and curl-driven upwelling between the three subregions. For instance, the central subregion has strong year-round coastal upwelling whereas upwelling tends to be strong only in the summer season in the northern subregion (Pelegri and Benazzouz, 2015). Similarly, the Ekman-driven upwelling

in the southern subregion is restricted to the winter and late fall (Pelegri and Benazzouz, 2015). There are distinct differences also with regard to the wind curl-driven upwelling between the three subregions. Indeed, the wind stress curl is predominantly downwelling-favorable in the northern and central subregions and upwelling-favorable in the southern subregion. This enhances upwelling in the southern region, particularly in the open ocean, but its effect is only partially sampled in the present study given our focus on the coastal region. Other inter-regional variations may also stem from the design of the experiment. For

instance, our experiment doesn't identify coastal upwelling immediately south of Cape Blanc and between Capes Barbas and Bojador because their bathymetry is shallower than the 70 m upwelling depth criterion used here . Furthermore, the upwelling strip is relatively narrow in the northern subregion because of the narrower shelf, thus limiting the offshore spread of coastal upwelling there (Figure 4 in SI).

## 3.2 Upwelling of nitrogen

The upwelling patterns of water and nitrogen have a few important differences. The southern subregion has the lowest upwelling water volume yet the strongest upwelling flux of nitrogen (Fig. 6). This is primarily due to the shallower nutricline associated with a shallower thermocline there as well as the higher-nutrient content of the South Atlantic Central Waters (SACW) that feed upwelling south of Cape Blanc. The central subregion has a moderate nitrogen flux associated with upwelling while the northern subregion has the weakest upwelling flux of nitrogen (Table 1, Fig. 6). Finally, parcels that upwell further from the

coast carry a nitrogen flux that is disproportionately lower than their corresponding associated water volume.



**Figure 6.** (a) Zonally integrated annual upwelling nitrogen. (b,c,d) Annual upwelling nitrogen with distance from the coast in the (b) southern, (c) central and (d) northern subregions. Light and dark blue segments represent contribution by deep and shallow sources of water parcels, respectively.



## 4 Offshore transport

### 4.1 Reach and timescales of offshore transport

Over the whole CanCS, the water offshore transit times are longer in the nearshore region and decrease as we go further away from the coast (Table 2). Large differences in the offshore transport timescales exist between the three CanCS subregions. In
the first 200 km from the coast, the offshore transport is fastest in the central subregion and slowest in the northern subregion (Table 2 and Fig. 5, SI). Indeed, in the central subregion, 80% of the upwelling particles reach 200 km in three months and 50% reach that distance in two months only. In contrast, less than 30% of particles upwelling in the northern subregion reach 200 km offshore in three months and it takes more than four months for half of them to reach that distance. At larger distances from the coast (beyond 400 km), the offshore transport becomes faster in the southern subregion with nearly 80% of particles
reaching 1200 km in two years, while only 70% and 30% of particles upwelling in the central and northern subregions, reach that distance in two years. This can also be seen in the water residence times being shortest in the central subregion up to 400 km and in the southern subregion beyond 400 km (Table 1, SI).

| | Upwell-200km | 200-400km | 400-600km | 600-800km | 800-1000km | 1000-1200km |
|---|---|---|---|---|---|---|
| Southern | 84 | 43 | 38 | 36 | 41 | 36 |
| Central | 59 | 60 | 56 | 50 | 45 | 43 |
| Northern | 126 | 96 | 72 | 56 | 44 | 42 |
| Whole CanCS | 83 | 66 | 57 | 49 | 44 | 41 |

**Table 2.** The median transit time (days) for particles to traverse each distance interval for the first time for the three subregions as well as the entire CanCS region in our experiment. The first column shows the transit time between upwelling and reaching 200 km offshore. Only particles that traverse the whole distance range are considered.

### 4.2 Net offshore transport

Integrated over the whole analysis domain from 10°N to 41.5°N, the CanCS exports over 70'000 km$^3$ yr$^{-1}$ of water toward
the open North Atlantic (Figure 7a). The maximum offshore transport is reached at close to 150 km from the shore at around the edge of the coastal upwelling area. The transport decreases thereafter gradually as the number of particles reaching farther offshore distances within the two-year integration period declines. But even at a distance of 1200 km from the coast, the offshore transport still amounts to 40'000 km$^3$ yr$^{-1}$.

There are substantial differences between the different subregions. The central subregion of the Canary system is responsible
for over half of the entire net offshore transport of water from the CanCS at any distance from the coast (Figure 7a). The offshore transport of the northern subregion is initially very strong, but decreases sharply thereafter, so that beyond 800 km, it becomes the weakest of all three regions. This is because of its low offshore transport efficiency and the relatively low volume



of upwelling (Table 1, Figure 7a). The southern subregion is associated with the smallest offshore transport of water up to 800 km offshore due to its small coastal upwelling volume.

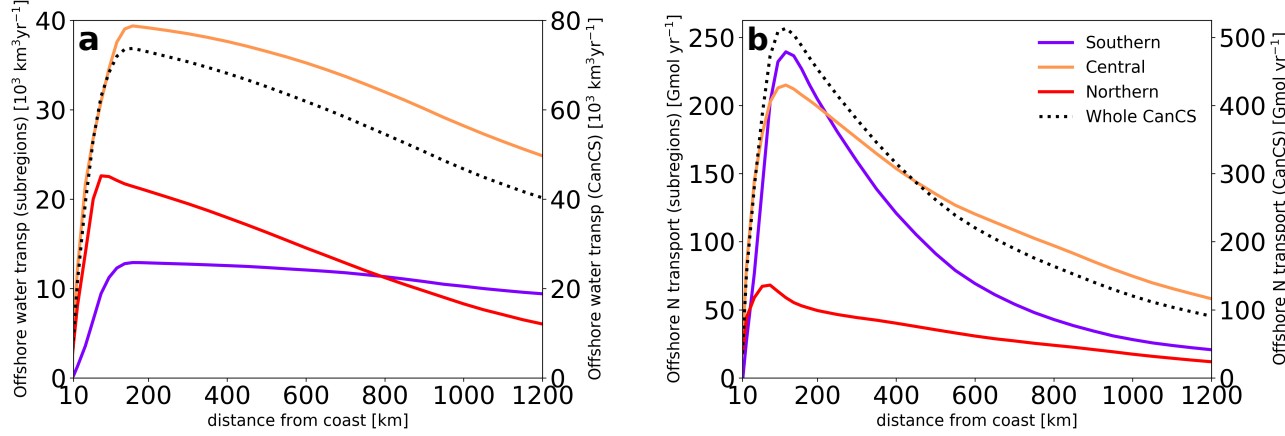

**Figure 7.** Net offshore transport of (a) water and (b) nitrogen as a function of the distance to the coast in the three subregions (left axis) as well as the entire CanCS (right axis) in our experiment.

At close to 150 km from the coast (corresponding to the edge of the coastal upwelling stripe), the offshore transport of nitrogen reaches its maximum, reaching values as large as 500 Gmol yr$^{-1}$. Thereafter, the offshore transport decreases expo-
nentially, although with a relatively long decay length scale, such that at a distance of 1200 km, the offshore export of nitrogen still amounts to 100 Gmol yr$^{-1}$.

In spite of having the lowest offshore transport of water, the southern subregion exports the highest amount of nitrogen offshore at 200 km (nearly half of the offshore export of nitrogen by the whole CanCS region at this distance). This is a direct consequence of the large upwelling flux of nitrogen (Table 1 and Fig. 7b). At larger distances from the coast, the situation reverses. Beyond 200 km from the coast, the offshore transport of nitrogen associated with the central subregion exceeds that originating from both the southern and northern subregions. At all distances from coast past 600km, the central subregion contributes over half of the nitrogen offshore export by the entire CanCS. At these distances, the nitrogen transport of waters stemming from the central subregion is at least twice as large as that from the southern subregion, and four times as large as that from the northern subregion.

The magnitude of the offshore transport of nitrogen at any distance from the coast depends both on the volume of offshore transport and how efficiently nitrogen is stripped from the waters by biological productivity and the resulting organic nitrogen exported to depth. It is thus instructive to assess the specific forms of nitrogen being transported offshore.

## 4.3 Nitrogen allocation

In the nearshore 50 km, phytoplankton is very efficient in taking up the inorganic nitrogen that is being upwelled, and fixing it into organic forms of nitrogen (Fig 8a). This results in nearly 100% of the offshore transported nitrogen to be in the form





**Figure 8.** The net offshore transport of each pool of nitrogen (phytoplankton, zooplankton, small and large detritus, nitrate and ammonium) as a function of the distance to the coast in the three subregions (left axis). The fraction of organic nitrogen (in %) in the net offshore transport of nitrogen at each distance from the coast is shown on the right axis.




of organic nitrogen. But as additional nitrogen is being supplied from below, the fraction of the upwelled nitrogen that gets consumed decreases rapidly with increasing offshore distance. Furthermore, some of the fixed organic nitrogen is being lost through sinking, so that beyond the nearshore 50 km region, inorganic nitrogen in the form of nitrate dominates the nitrogen pool at all distances from the coast (Fig 8a). Beyond 200 km, ammonia is the second largest pool. Overall, organic nitrogen

contributes only 30% to the total nitrogen pool. Within the organic nitrogen pool, small detritus contributes the most to the offshore transport, while the contribution of phyto- and zooplankton is much smaller and that of the large detritus particles essentially negligible.

Given its dominance in terms of the total offshore transport, the southern subdomain is also the main region determining the whole CanCS pattern of nitrogen allocation (Fig 8b). In this southern subdomain, the fraction of organic nitrogen is particularly

low, being only 20% at the peak of the offshore transport. With increasing distance, the fraction increases to 30%. This indicates a much further offshore extension of the conversion of inorganic nutrients to organic matter in this domain. The central domain has a nitrogen allocation pattern that is similar to that of the whole CanCS (Fig 8c), while the transport in the northern subdomain is not only weak, but also the least dominated by the offshore transport of nitrate (Fig 8d).

The different offshore gradients exhibited by the different pools of nitrogen can be largely explained by their position in the

cycling of nitrogen within the euphotic zone and their susceptibility to sinking. Nitrate, being the dominant form of nitrogen being upwelled comes first, followed by phytoplankton, zooplankton, and small detritus. The latter two contribute then to the formation of ammonia by respiration and remineralization. Ammonia tends to accumulate, partially aided by its non-sinking nature, making this an important part of the offshore transport. In contrast, the very small contribution by the large detritus is largely a consequence of its rapid export to depth (see also Gruber et al., 2006).

## 4.4 Depth structure of offshore transport

Upwelling particles that are transported offshore are also subject to vertical circulation that distributes them vertically (Figs 7-8 in SI). For the northern and central subregions, upwelling particles are subject to moderate subduction at around 100 to 200 km from the coast (Fig. 7, 8c, SI). This subduction is particularly strong in the upper 70 m and vanishes around 200 m. In the southern subregion, however, upwelling particles are subject to very little subduction while a strong secondary upwelling in the

open ocean maintains them near the surface (Fig. 10 and Fig. 8c, SI). Indeed, the vast majority (>90%) of particles upwelling in the southern subregion remain in the upper 100 m as they are transported offshore. In the central and northern subregions a smaller (>80%) but still important proportion of upwelling particles remain in the upper 100 m as they are exported offshore (Fig. 9 and Fig. 8, SI). The stronger downward advective transport occurring in the central and northern subregions is due to a strong negative wind stress curl in these subregions, which is absent in the southern subregion (Lovecchio et al., 2017).

Furthermore, persistent filaments associated with prominent capes in the central and northern subregions may contribute to enhanced subduction of upwelling water and nutrients there. Similarly to the water volume transport, subduction of nitrogen is strongest in the northern subregion and weakest in the southern subregion (Fig. 10). For instance, at 400 km above 95% of nitrogen upwelled in the southern subregion remains in the top 100 m. This proportion drops to around 80% and 60% for the central and northern subregions, respectively (Fig. 9b).



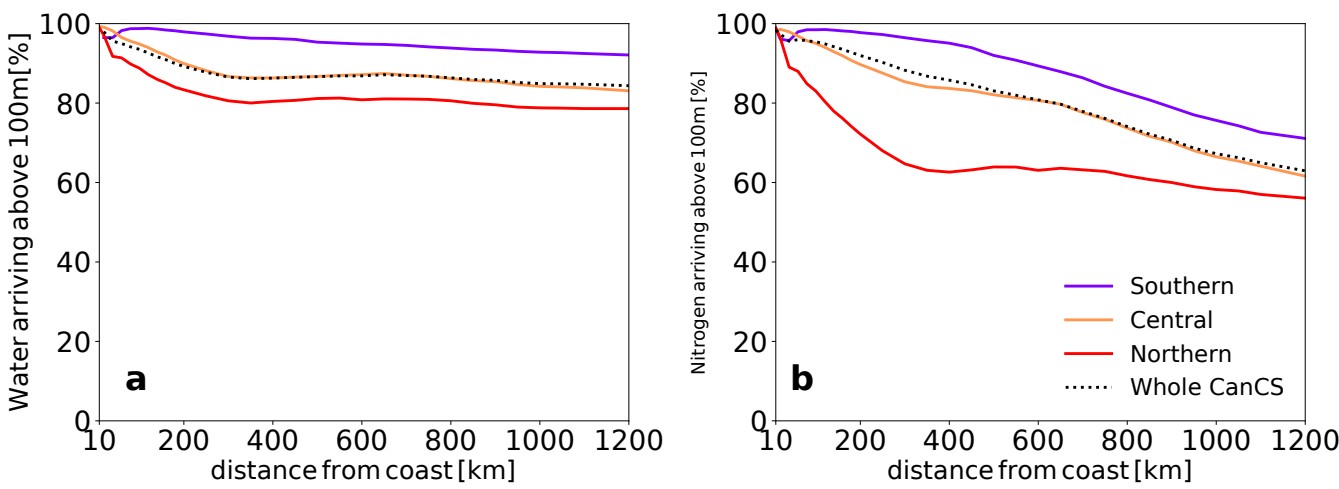

**Figure 9.** Fraction (in %) of upwelled (a) water and (b) nitrogen that remains above 100 m at their first arrival to each offshore distance in the three subregions as well as the entire CanCS region in our experiment.







**Figure 10.** Meridionally integrated net downward and offshore transports of organic (left) and inorganic (right) nitrogen in the upper 200 m as a function of the distance to the coast in the three subregions as well as the entire CanCS in our experiment. Color shading shows the downward transport (in Gmol N km$^{-1}$ yr$^{-1}$) while contours correspond to the offshore transport (in Gmol N m$^{-1}$ yr$^{-1}$).



Finally, the offshore transport of organic nitrogen is smaller in magnitude than that of inorganic nitrogen and is mostly limited to the near surface in all subregions (Fig. 10 and Fig. 9, SI). In contrast, the transport of inorganic nitrogen shows a subsurface secondary maximum at between 50 and 100 m in addition to the surface maximum (Fig. 10).

## 4.5 Horizontal structure of offshore transport

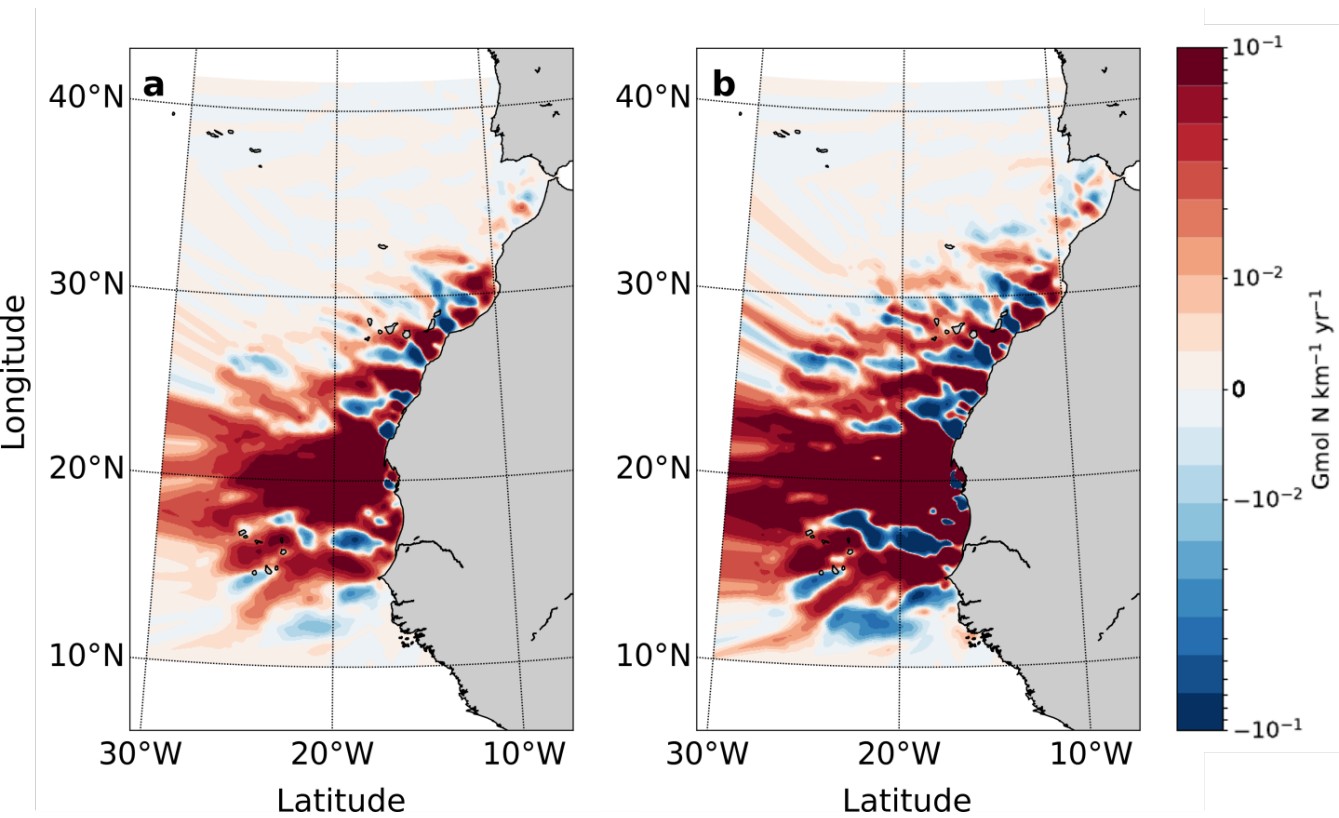

**Figure 11.** Horizontal distribution of vertically integrated offshore transport of organic (a) and inorganic (b) nitrogen for the whole CanCS region in our experiment.

5    The convergence of the Canary coastal current flowing from the north and the Mauritanian current flowing from the south leads to a strong offshore transport of nitrogen around cape Blanc (21°N) (Fig. 11 and Fig. 10, SI). The nitrogen (both organic and inorganic) channelled through the confluence of the two currents originates predominantly from waters upwelling in the central and southern subregions, with a small contribution from particles upwelling in the northern subregion. This is consistent with the finding of Lovecchio et al. (2017) who - using a Eulerian approach - also found the Cape Verde Front

10    to be a major channel of export of upwelled waters and of the associated nitrogen into the open ocean. Although integrated vertically and averaged in time, the pattern of offshore transport captures a spatially averaged manifestation of mesoscale eddies in latitudinally alternating offshore-onshore corridors known as striations. The emergence of striations has been described in





Davis et al. (2014). The pattern also captures the role of filaments in the enhancement of transport at certain capes where there is spatial consistency in coastal filament formation (more in Section 4.1). The presence of striations in offshore flow and the role of filaments in offshore transport have also been documented in Lovecchio et al. (2017), and the latter has been studied in detail in Lovecchio et al. (2018).

## 5 Mechanisms of transport

### 5.1 Enhancement of offshore transport around capes

The coastal upwelling front associated with the strong density gradient formed at the transition zone between the upwelling waters and the open ocean waters limits the offshore transport. Yet, the cold coastal filaments that emerge from the instability of the upwelling front and its interaction with eddies can play a crucial role in transporting upwelling waters across the front against the mean density gradient. These filaments have been shown to be associated with a high export of organic material and nutrients into the open ocean, particularly in the nearshore. Àlvarez-Salgado et al. (2007) suggested the contribution of filaments to the transport of carbon off Iberia and NW Africa to be 2.5 to 4.5 times larger than the export driven by Ekman transport. Lovecchio et al. (2018) estimate that coastal filaments are responsible for 80% of the offshore transport in the first 100 km offshore. Coastal filaments along the West African coast can occur everywhere, but it is well established that the majority of the filaments are persistently associated with the major capes along the coast. Lovecchio et al. (2018) demonstrated that in their analyses, 25% of the time, the filaments were associated with capes. Many filaments originating at capes are so persistent and well known that they have been named after the cape, such as the Cape Ghir filament (Pelegri et al., 2005b) and the exceptionally large and extensive Cape Blanc/Barbas filament that occurs at the confluence of the Canary and Mauritanian currents (van Camp et al., 1991; Barton et al., 2004). Enhanced transport can also be seen at other capes such as Cape Bojador where a persistent filament extending up to 500km offshore can be observed (Figure not shown).

Here, we explore how areas around the major capes that are favorable to strong filamentary activity affect upwelling and offshore export of water and nitrogen. These are Capes Verde (14.5°N - 15.5°N) and Blanc (20°N - 21°N) in the southern subregion, Capes Barbas (22°N - 22.75°N), Bojador (25°N - 26.5°N) and Juby (27.5°N - 28.5°N) in the central subregion and Capes Dra (28.5°N - 29.5°N) and Ghir (30°N - 31°N) in the northern subregion (Fig. 4a and Table 2, SI). Previous studies of the Canary upwelling mostly stressed the role of filaments in locally enhancing offshore transport regardless of the source of upwelling. Similarly, we find an enhancement of offshore transport of upwelling particles in the first 200km from the coast within the latitudinal range of all capes except Capes Verde and Juby (Figure 12). Transport at latitudes of capes Blanc, Barbas, Dra and Ghir is larger than at the non-cape latitudes as far as 1000 km from the coast. We further separately consider the enhancement of offshore transport associated with (i) enhanced local upwelling around the capes and (ii) increased export of remotely upwelled water at each cape, i.e., the export of waters that upwelled far away from the capes but is then transported along the coast toward the cape, from where it is exported toward the open ocean.

The offshore transport of water upwelling around Cape Bojador in the central subregion and Capes Dra and Ghir in the northern subregion is 20% to 30% larger than that originating from non-cape areas (per coastal length) (Figure 11 in SI).





Corresponding enhancement in local nitrogen upwelling and export is seen only in Capes Dra and Ghir out of the seven capes examined (Figure 12). Cape Bojador does not show enhanced local nitrogen upwelling (although a slight enhancement is shown in its offshore export).



**Figure 12.** Enhancement of net offshore transport of nitrogen by capes. (a,b,c) Transport within the latitudinal span of each cape or non-cape area. Transport occurring within each latitudinal span is considered at any distance from the coast irrespective of the location of upwelling or coastal export. (d,e,f) Transport by water upwelling locally in each cape or non-cape coast. (g,h,i) Transport by particles that leave the coastal upwelling region at each cape or non-cape area but upwell remotely. Note that the transport is normalized by coastal length (divided by the length of the coast at the respective cape/non-cape).

Cape Blanc, Cape Barbas and Cape Ghir are highly efficient in exporting remotely upwelled waters to the open ocean (Fig. 12, SI). Cape Blanc's enhancement of offshore export of nitrogen by remote upwelling compared to the non-cape part of the central subregion is 350% - 400% in the first 400 km then constantly increases further offshore to reach a 935% enhancement at 1200 km. Similarly, Cape Barbas shows an enhancement of remote upwelling of about 200 % for all distances from the





coast while Cape Ghir shows a 250% enhancement from 100 km to 800 km offshore (Figure 12). The offshore transport of nitrogen by remote upwelling exported by all capes constitutes more than 30% of the total offshore transport of upwelling at all distances from the coast (Fig. 7 and Fig. 12). In fact, all capes source the majority of water and nitrogen they export from remote upwelling (Table 3). All capes also source more of their export from remote upwelling compared to the rest of the

5   coast. Remote upwelling accounts for over 75% of the nitrogen export in all capes except Cape Verde and Cape Bojador while it accounts for a minority of the source of non-cape parts of the coast except in the central subregion (Table 3).

The analysis of the source waters for particles exported around the major capes reveals a strong alongshore transport of upwelled particles that connects upwelling between different coastal regions (Fig. 13). This is consistent with previous studies that found strong meridional alongshore advection of nutrients (Carr and Kearns, 2003; Pelegri et al., 2006; Pastor et al.,

10  2013; Pelegri and Benazzouz, 2015; Auger et al., 2016; Lovecchio et al., 2017). With the exception of Cape Blanc and Cape Barbas, most capes export offshore water that mostly first upwells north of their latitude (Fig. 13). In the central and northern subregions, this is primarily due to the southward offshore flow by the Canary current. In the southern subregion, the effect of the Mauritanian current is visible in the southern source of upwelling that leaves the coast at Cape Blanc and Cape Barbas that act to concentrate remote upwelling from both the north and the south and channel it offshore (Figure 13). We conclude

15  that capes such as Cape Blanc and Cape Barbas act to concentrate and export remotely upwelled waters, and that this effect is much more important than the occurrence of enhanced local upwelling at the capes.

| Cape/Non-cape | Local N (Mmol N km$^{-1}$) | Remote N (Mmol N km$^{-1}$) | Local N (%) | Local Water (%) |
|---|---|---|---|---|
| Verde | 12.5 | 32.8 | 27.6 | 21.7 |
| Blanc | 35.1 | 233.0 | 13.1 | 11.5 |
| Non-cape (southern) | 393.7 | 220.7 | 64.1 | 59.3 |
| Barbas | 8.9 | 83.3 | 9.6 | 8.0 |
| Bojador | 17.7 | 41.7 | 29.8 | 26.5 |
| Juby | 2.4 | 14.6 | 13.9 | 10.5 |
| Non-cape (central) | 115.1 | 200.7 | 36.5 | 33.5 |
| Dra | 1.1 | 5.4 | 17.1 | 14.1 |
| Ghir | 3.1 | 11.9 | 20.7 | 20.3 |
| Non-cape (northern) | 12.5 | 2.6 | 82.7 | 87.3 |

**Table 3.** First two columns show daily contribution of local and remote upwelling to nitrogen that leaves the coastal region at each cape (in Mmol N km$^{-1}$) per km of coastal length. The last two columns show the contribution of local upwelling (in %) to total nitrogen and water exported at each cape or non-cape coast.



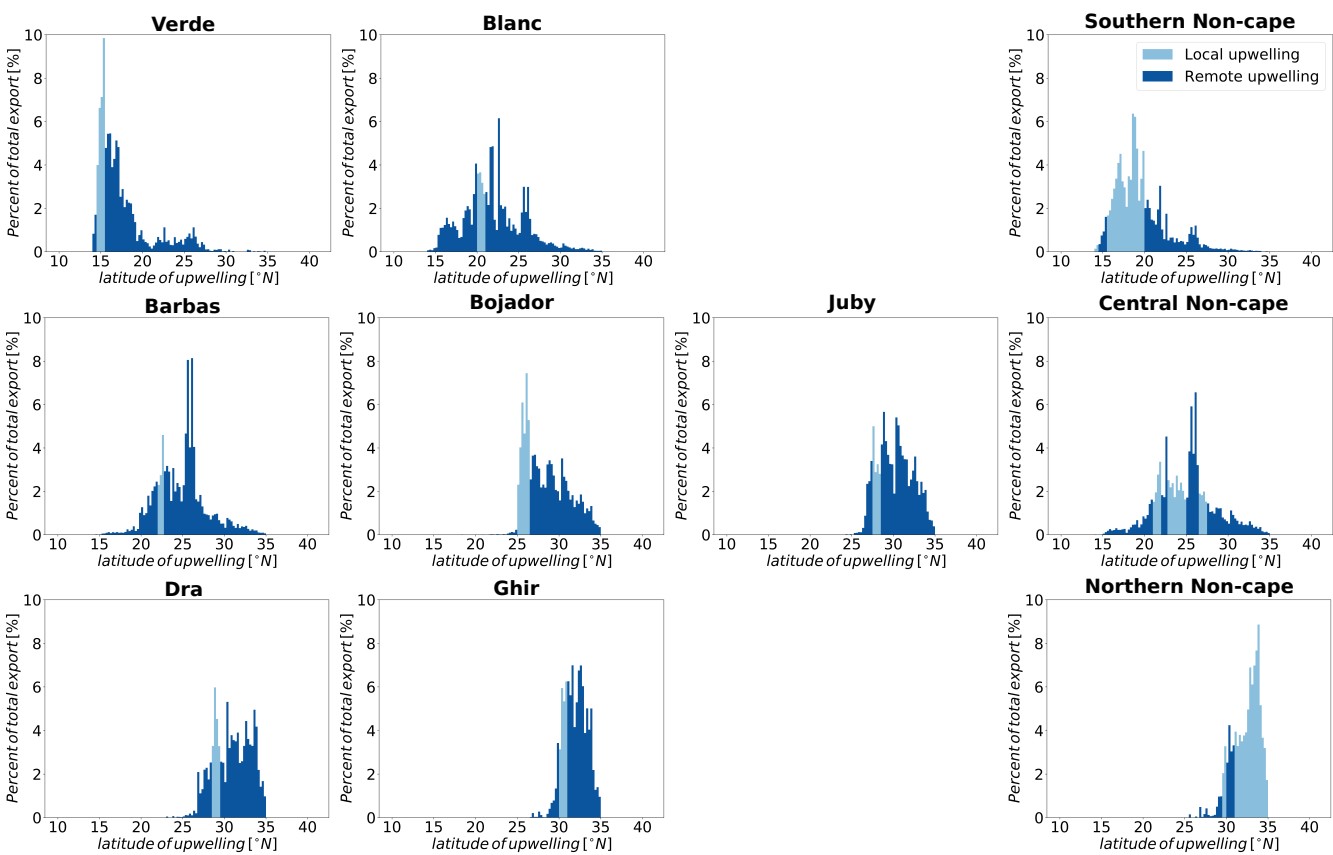

**Figure 13.** Latitude of upwelling of nitrogen exported at each cape or non-cape coast.

## 5.2 Role of recirculation

Particles that are transported offshore past a certain distance can return back to that same distance or never cross it again. We term the transport associated with the latter "direct transport". Particles that cross the same distance multiple times can end up further offshore (an odd number of crossings) or closer to the coast (an even number of crossings). We term them "indirect transport" and "net recirculation", respectively (Figure 14a).

It is worth noting that only direct and indirect transport contribute to the net offshore transport of water, while net recirculation does not (although it may slightly contribute to transport of nitrogen) (Figure 14). The sum of direct and indirect transports adds up to give the net transport of water to each distance while the sum of all three represents the total volume of water that has reached each distance at any point during the experiment. In the nearshore region, the meandering of the Canary Current as well as the coastal upwelling cell can cause upwelling particles to recirculate closer to the coast (Mittelstaedt and Hamann, 1981; Mittelstaedt, 1983; Estrade et al., 2008). But the leading cause of recirculation in the open ocean is the ubiquitious pres-

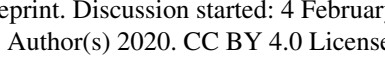



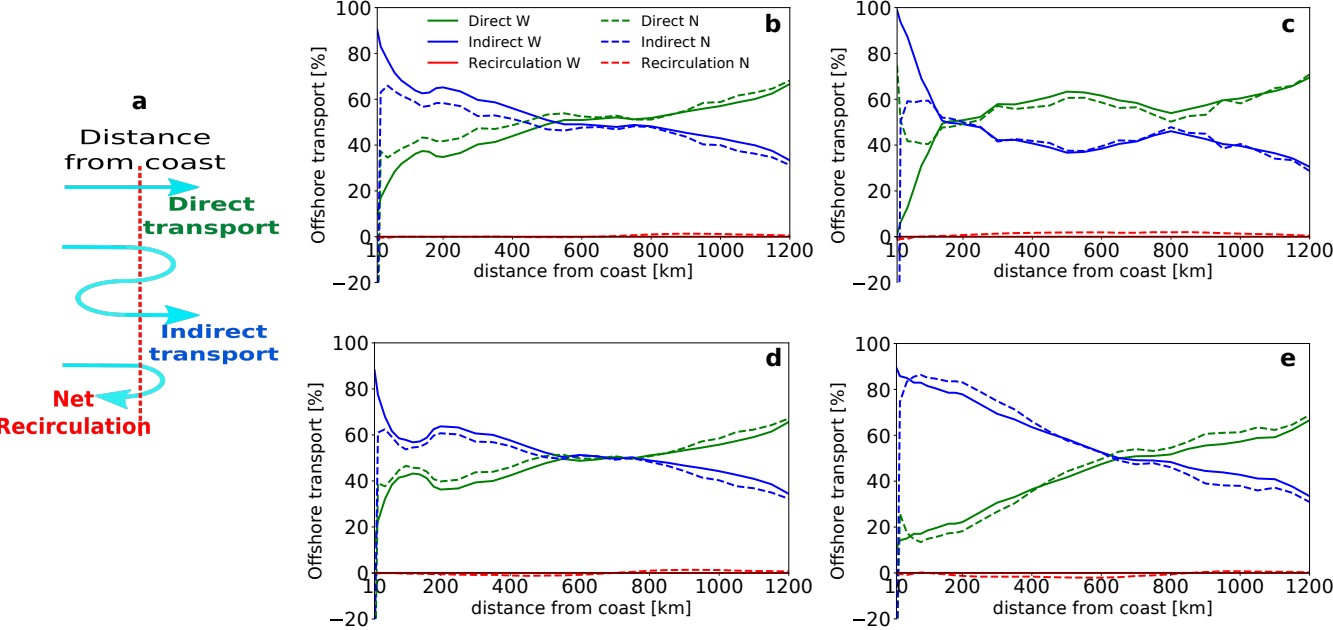

**Figure 14.** (a) Schematic showing the three types of offshore transport trajectories. (b,c,d,e) Contribution (in %) of direct transport (green), indirect transport (blue) and net recirculation (red) to the total net offshore transport of water (solid) and nitrogen (dashed) at each distance to the coast in the entire CanCS region (b) as well as the southern (c), central (d) and northern (e) subregions. Note that the net recirculation has no contribution to net offshore water transport but contributes slightly to the net offshore transport of nitrogen.

ence of mesoscale structures, particularly eddies. Therefore, contrasting the direct and indirect components of the transport can be used to gauge the relative importance of the mesoscale eddies and current meandering in the offshore transport.

Examining the three components of the offshore transport in the three subregions reveals that the direct and indirect components virtually explain all the transport of water and nitrogen at all latitudes and distances from the coast. Yet, important differences exist between the three subregions in terms of the relative importance of these two forms of transport (Fig. 14).

5 In the southern subregion, the direct transport contributes by a large share to the total transport of water and nitrogen at all distances except in the first 200 km offshore where the indirect transport dominates. In contrast, in the central and northern subregions, the direct transport has a greater share to transport only past 700 km offshore, while the indirect transport dominates closer to the coast. The relative importance of the indirect transport is particularly strong between 100 km and 400 km in these two subregions. For instance, at 200 km the indirect transport contributes by up to 65% and 80% to the total offshore transport

10 of water in the central and northern subregions, respectively. For the transport of nitrogen, the share of the indirect transport at the same distance is 60% and 85% in the two subregions.

The importance of the indirect transport in the central and northern subregions can be linked to the prominent role played by mesoscale eddies there. Indeed, eddies with length scale of 100 km to 300 km are known to be important at these latitudes





(Mittelstaedt, 1991). These include a recurrent cyclonic eddy south of Cape Juby and the cyclonic and anticyclonic eddies entrained by the Canary Archipelago, forming the so called Canary Eddy Corridor (CEC), which is located at $22° - 29°$ (Arístegui et al., 1994; Piedeleu et al., 2009; Sangrà et al. 2009). This region of long-lived westward-propagating eddies is known to contribute strongly to the offshore transport of organic matter and carbon (Sangrà et al., 2009).

The prominence of water recirculation in the central and northern subregions - materialized by the dominance of the indirect transport - up to 600-800 km offshore is consistent with the longer crossshore transit times characterizing these latitudes (Table 2 and Fig. 13, SI). This suggests that recirculation acts to slow down the offshore transport and may explain the less efficient offshore transport of water upwelling from these regions (Fig. 7a).

## 6 Potential contribution to open ocean nitrogen budget

The strong export of nutrients and organic matter from the CanUS to the oligotrophic open ocean fuels new production (NP) and contributes to heterotrophy there. Here we quantify this contribution in the North Atlantic Tropical Gyral Province (NATR) and the North Atlantic Subtropical Gyral East (NASE) provinces as defined by Longhurst et al. (2007) (see Fig. 3 in SI). The Lagrangian approach allows for the isolating of the contribution of the upwelling particles to the transport of nitrogen into the NATR and NASE provinces, adjacent to the CUS (see Fig. 3, SI).

NP is estimated from the Net Primary Production (NPP) and the available estimates of f-ratio (f-ratio$= \frac{NP}{NPP}$) in the literature. NPP is derived in each province from the Vertically Generalized Production Model (VGPM) using sea color data from 1997 to 2017 (Table 4). We use estimates of the f-ratio from Laws et al. (2000) (see their Plate 3) as well as previous estimates of the export efficiency (i.e., e-factor) from Henson et al. (2011) and Siegel et al. (2014) since the two are equivalent for sufficiently large open ocean regions (Eppley and Peterson, 1979). The derived f-ratio values are typically low and vary from 4% in the NATR according to Henson et al. (2011) to 15% in the NASE following Laws et al. (2000) (Table 4).

The southern, central and northern coastal upwelling subregions in our study have total upwellings of 282.3, 200.9 and 79.2 Gigamol N yr$^{-1}$, respectively. From this total of 562.4 Gmol N yr$^{-1}$, 357.6 Gmol N yr$^{-1}$ and 32.4 Gmol N yr$^{-1}$ reach the upper 100m of the NATR and the NASE, respectively. When normalized by the area of each province, the CanCS upwelling source corresponds to 43.7 mmol N m$^{-2}$ yr$^{-1}$ and 7.3 mmol N m$^{-2}$ yr$^{-1}$ for the NATR and NASE, respectively (Table 4).

Most of the upwelled nitrogen exported to The NATR originates from the central (46.5%) and southern subregions (43%), whereas the nitrogen upwelled in the northern subregion dominates (76.7%) the nitrogen supplied to the NASE. Hence, the nitrogen transported (in the top 100m) from the Canary coastal upwelling to the NATR and NASE provinces represents 31.2 - 62.3 % and 2.1 - 4.9% of their total NP, respectively, based on NP values calculated from Laws et al. (2000), Siegel et al. (2014) and Henson et al. (2011).

Finally, the contribution of the Canary upwelling also appears significant in comparison to other major sources of new nitrogen to the oligotrophic open North Atlantic Ocean such as N$_2$ fixation (Moore et al, 2009; Luo et al., 2012; Fernandez-Castro et al., 2015), atmospheric deposition (Duce et al., 2008) and net meridional transfer through Ekman divergence (Williams and Follows, 1998). Indeed, the upwelling source seems to exceed contributions from N$_2$ fixation, the atmospheric deposition and





**Table 4.** Nitrogen supply to the NATR and NASE provinces (in mmol N m$^{-2}$ yr$^{-1}$) by the Canary upwelling and other sources and the estimated new production (NP) in the two provinces. Contribution of upwelling by each subregion is calculated by adding the net amount of nitrogen upwelling particles carry to each province in the top 100 m by the end of the experiment.

| N source | NATR | NASE |
|---|---|---|
| Canary upwelling | | |
| Deep source | | |
| 14-21°N | 14.5 | > 0.0 |
| 21-28°N | 18.2 | 1.5 |
| 28-35°N | 3.8 | 4.8 |
| Total | 36.5 | 6.3 |
| Shallow source | | |
| 14-21°N | 4.3 | > 0.0 |
| 21-28°N | 2.1 | 0.1 |
| 28-35°N | 0.8 | 0.8 |
| Total | 7.2 | 1.0 |
| NPP (VGPM) | 1753 | 2139 |
| NP | | |
| Laws et al. (2000)* (e-factor=0.06, 0.16) | 105.2 | 342.2 |
| Siegel et al. (2014)* (e-factor=0.08 for both) | 140.2 | 171.1 |
| Henson et al. (2011)* (e-factor=0.04, 0.07) | 70.1 | 149.7 |
| N$_2$ biological fixation | | |
| Luo et al. (2012) | 22.5 ± 12.5 | 1.5 ± 0.4 |
| Moore et al. (2009)* | 23 | < 3 |
| Fernández-Castro et al. (2015) | 3.1 ± 2.4 | 2.5 ± 1.1 |
| Atmospheric deposition (Duce et al., 2008) | 9.5 ± 2.5 | 14.5 ± 2.5 |
| Net meridional transfer (Williams and Follows, 1998) | 30 (NO$_3$ only) | 30 (NO$_3$ only) |

\* estimates are based on maps provided by the cited paper

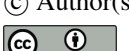



meridional transfer in the NATR province. In the NASE province, the upwelling contribution is larger than the $N_2$ fixation but weaker than the atmospheric deposition and the meridional transfer (Table 4).

## 7 Comparison with previous studies

**Table 5.** Comparison of POC offshore transport estimates in the CanCS region from previous studies and from the present study. Note that our model-based estimates correspond to annual mean climatologies whereas observation-based estimates listed in the table are based on data collected from different individual campaigns that took place either in August or September. All values are given in kg C s$^{-1}$ with a C:N ratio of 6.625 assumed in our model.

| Previous study | Latitude | Longitude (offshore dist.) | Depth | Time | Transport estimate | This study |
|---|---|---|---|---|---|---|
| Alonso-González et al. (2009) | 20°N-29.2°N | 20.35°W | 0m-73m | 7-29 Sep, 2003 | 161.3 | 109.8 |
| | - | 26°W | 0m-135m | - | 212.7 | 40.3 |
| García-Muñoz et al. (2004) | 27.6°N-27.85°N | 13.7°W | 0m-100m | 13-26 Aug, 1999 | 18.8 | 10.6 |
| García-Muñoz et al. (2005) | 30.8°N-32.4°N | 11.25°W | 0m-100m | 26 Sep-2 Oct, 1997 | 11.2 | 5.6 |
| Santana-Falcon et al. (2016) | 30.2°N-31.2°N | 10.6°W | 0m-100m | 26-27 Aug, 2009 | 20.9 | 3.4 |
| Lovecchio et al. (2017) | 17°N-24.5°N | 100km | 0m-100m | Annual mean | 404 | 273.3 |
| | - | 500 km | - | - | 207 | 176.8 |
| | - | 1000 km | - | - | 70 | 61.1 |
| | 24.5°N-32°N | 100km | - | - | 198 | 93.2 |
| | - | 500 km | - | - | 34 | 16.1 |
| | - | 1000 km | - | - | 11 | 3.2 |

Several previous studies of the CanCS estimated the offshore transport of particulate organic carbon (POC) at various locations off the Northwest African coast (Table 5). For instance, Alonso-González et al. (2009) derived estimates of offshore transport of suspended POC in the upper ocean in a box model approach using in-situ measurements of POC and velocities based on geostrophy and Ekman contribution computed from remotely sensed estimates. The POC export was quantified between 20°N and 29.2°N at two different transects at 20.6° W and 26° W for the top water column with a density of 26.44 kg m$^{-3}$ or less. Their transport estimates exceed ours by around 45% and a factor five at 20.6° W and 26° W, respectively (Table 5). These discrepancies might result from the important sampling differences between the two studies, as our estimates correspond to annual-mean climatologies and are restricted to waters that upwell near the coast in contrast to the observational study. We also note that the mismatch between the two studies is particularly large for the deeper (0-135 m) and the further offshore (26° W) transect that is more likely to be affected by waters not directly related to coastal upwelling. García-Muñoz et al. (2004) and García-Muñoz et al. (2005) estimated the offshore transport of POC in the top 100 m associated with two filaments near C. Juby and C. Ghir, respectively. These estimates are nearly twice as large as ours (Table 5). Similarly, Santana-Falcon et al. (2016) inferred the offshore transport of POC associated with a filament near C. Ghir observed in August 2009





(Table 5). Their estimate exceeds our model-based annual-mean offshore transport estimate by almost a factor six (Table 5). Finally, we contrast annual mean POC offshore transport estimates by Lovecchio et al. (2017) to ours. While the transport figures by the two studies are relatively close between 17°N and 24.5°N at large offshore distances (> 500 km), our estimated transport is nearly twice weaker in the nearshore and up to three times smaller at 1000 km between 24.5°N and 32°N (Table

5). While Lovecchio et al. (2017) is the only previous study that provides model-based large-scale annual mean estimates of the offshore transport of POC in the CanCS, it is based on Eulerian approach that does not distinguish between waters that originate from the coastal upwelling and the non-upwelling waters.

In summary, all previous studies reported here find higher estimates of POC offshore export in several portions of the CanCS. Our lower offshore transport estimates may at least partially result from our Lagrangian approach that focuses on coastally

upwelled water only, disregarding non-upwelling waters and open ocean upwelling. Furthermore, restricting the analysis to upwelling particles in our study limits offshore flux to near-surface waters as most upwelled waters remain at very shallow depth. Finally, the loss of organic nitrogen due to sinking of particulate matter experienced by upwelling waters as they are exported offshore likely contributes to our overall lower transport estimates.

Previous model-based studies of the Benguela and California coastal upwelling systems suggest other EBUS may also

significantly contribute to the nitrogen budget of adjacent open oceans (Table 4, SI). For instance, Gutknecht et al. (2013) found significant net offshore transport of nitrogen in the Benguela current system (BenCS), with organic nitrogen exceeding inorganic nitrogen at far offshore distances. They estimated the contribution of the BenCS in the top 50m, to amount to $100 \pm 40$ mmol N m$^{-2}$ yr$^{-1}$ to the adjacent South Atlantic Subtropical Gyral Province according to the classification by Longhurst (2007) (Gutknecht et al., 2013). This contribution, however, is likely overestimated since results in the highly productive Walvis

Bay area (22°S-24°S) were linearly extrapolated to the whole BenCS. Frischknecht et al. (2018) also found high offshore export of both organic and inorganic nitrogen from the productive central region of the California current system (CalCS) to the open ocean as far as 500 km from the coast (Table 4, SI). The efficiency of offshore transport of organic matter and dissolved inorganic nitrogen in the top 100 m between 100 km and 500 km are almost identical between their study of the CalCS and the present study of the CanCS.

## 25   8   Implications and caveats

Our Lagrangian approach identified waters upwelling along the Northwest African coast and characterized the kinetics, structure and timescale of their offshore transport. Using a Lagrangian approach allowed us to assess the reach and fate of coastally upwelled nitrogen far from the coast as well as study its biogeochemical transformation along particle trajectories. Our estimate of offshore export of organic nitrogen/carbon by upwelling is lower than the estimate by Lovecchio et al. (2017), at all dis-

tances from the coast (Table 3 in SI). These differences can in particular stem from the differences in approaches (Lagrangian vs. Eulerian) and the studied parts of the CanCS (their budget is computed for the area between 9.5°N and 32°N) that differ between the two studies. We found the offshore transport to be generally slowest in the first few hundreds of km from the coast and to increase in speed with increasing offshore distances (Table 2). This is particularly true for the northern and the



southern subregions of the CanCS. The existence of an inverse relationship between upwelling water recirculation and their cross-shore transit times (Fig. 13, SI) suggests the slowdown of offshore transport in the first 300-500km to result from a larger role played by eddies that act to weaken the net offshore transport there. Previous studies have highlighted the role of eddies in contributing to organic carbon and nutrient offshore transport in EBUS (Nagai et al., 2015; Lovecchio et al., 2018). The

findings of the present study reveal that enhanced water recirculation driven by intense eddy activity may reduce the efficiency of the offshore transport of water.

In the first 200km, the offshore transport is fastest and the water recirculation is weakest in the central CanCS subregion (Table 2, Fig. 13, SI). We show that this region is characterized by the presence of major filament-generating capes such as Cape Blanc and Cape Barbas that contribute strongly to offshore transport of water and nutrients. They act as concentrators

that collect coastal water and nutrient and channel it to the open ocean as most of the water exported (84%) at the capes upwells remotely. Lovecchio et al. (2018) have highlighted the role of persistent filaments in mediating more than 80% of the offshore transport of organic carbon in the first 100km from the coast. Here, we show that key capes along the CanCS represent hotspots of offshore transport of water and nitrogen, thanks to the intense filaments that develop around them. Our findings also highlight the importance of the alongshore transport as most of the waters exported at capes have upwelled at latitudes far

away from the location of the capes. This is consistent with the concept of the three-dimensional biological pump highlighted in previous studies, according to which strong offshore and alongshore lateral transports decouple the locations of upwelling and of biological export (Plattner et al., 2005; Lovecchio et al. 2017, 2018).

As the along-trajectories nitrogen-depletion rates depend on the depth of the transport, the subduction of upwelling waters to deeper layers occurring mostly between 100 and 200km offshore tends to affect the efficiency of the offshore transport of

nitrogen. For instance, in the southern subregion where the transport is the shallowest, the nitrogen depletion is large (because of biological uptake and sinking), thus resulting in smaller supply of upwelled nitrogen to the North Atlantic gyre at large distances from the coast. Conversely, the northern and central subregions contribute relatively more to the open ocean nitrogen supply due to deeper transport and weaker nitrogen-depletion.

Finally, our study confirms the strong alongshore variations of the CanCS in terms of offshore transport reach and efficiency.

Our results show that the offshore transport of nitrogen off the CanCS depends on many important factors, including the intensity of upwelling, the efficiency of the offshore transport, controlled by the kinetics of the transport and the nitrogen depletion rates. In particular, factors such as eddies, filaments, alongshore transport all contribute to modulate the magnitude and reach of the water volume and nutrient content transported offshore. The complexity of these factors and their interactions explain the complexity of the offshore transport and its spatial diversity.

Yet, the study has several caveats that stem from the limitations of the experimental design or are inherent to the tools. For instance, our biogeochemical model is based on nitrogen and has no representation of other potentially limiting nutrients such as iron and phosphate. Furthermore, the model lacks a representation of nitrogen fixation. Yet, previous studies indicate nitrogen to be the main limiting nutrient (e.g., Moore et al., 2013) and nitrogen fixation to be very low (Luo et al., 2012; Moreira-Coello et al., 2017) in the CanCS upwelling region. Therefore, we believe these model limitations should not affect

the study's main conclusions. A potentially more serious limitation stems from our Ariane-based Lagrangian analysis that





includes resolved transport but does not explicitly take into account subgrid mixing in the computation of trajectories. This is a common problem in most Lagrangian trajectory based studies. We think including subgrid mixing should increase the vertical dispersion of upwelled water and nitrogen. While this may reduce the offshore transport efficiency (weaker offshore velocities at depth), it should also reduce along-trajectory nitrogen depletion. A compensation (even partial) between these two processes

may lead to an overall small change in the estimated total supply of upwelled nitrogen to the open North Atlantic ocean.

Finally, our coastal upwelling sampling involves subjective choices that can in theory affect our results. For instance, the identification of upwelling waters based on the upward crossing of a constant depth (70 m) is a somewhat arbitrary choice. Some previous Lagrangian studies have used cold sea surface temperature to identify and track upwelling waters (e.g. Rivas and Samelson, 2011). However, such definitions can be misleading as cold waters not necessarily associated with upwelling

may also be sampled. In the present study, our approach is based on the use of vertical velocities to identify upwelling. More specifically, we use the crossing of the 70 m depth as this corresponds to the average depth of the euphotic zone in the Canary coastal region. However, both upwelling depth and the depth of euphotic zone vary in space and time. Yet, Drake et al. (2018) show that defining upwelling with varying depth criteria yield results that are also well captured by experiments that use fixed depths. Furthermore, a variable upwelling depth definition can make it difficult to distinguish between upwelling variations

driven by the variability of atmospheric forcing and those caused by changes in the analysis depth.

It is worth noting that only coastal waters that originate from the open ocean are sampled as particles are released at the western section of the coastal region only. Therefore, water that may enter the control box along the coast from the north or the south is not considered. Excluding waters that enter along the shelf restricts our analysis to upwelling waters that carry deep ocean properties. Because of the much extended western section in comparison to the northern and southern exits, we believe

this potentially missing influx of coastal water to be small and unlikely to affect our results.

## 9   Summary and conclusions

We quantify the offshore export of water and nitrogen from the CanCS using a Lagrangian approach that consist in tracking all open ocean particles entering the coastal region and upwelling along the West African coast between 14°N and 35°N. Due to a larger upwelling volume and a faster offshore transport in the nearshore region (up to 400 km offshore), the central

CanCS subregion is responsible for the largest net offshore transport of water at any distance from the coast. Conversely, the southern CanCS is associated with the smallest offshore transport up to 800 km offshore due to a smaller volume of coastal upwelling. Yet, the southern subregion exports the highest amount of nitrogen offshore at 200 km because of a large upwelling flux of nitrogen associated with a shallower thermocline and higher subsurface nitrogen concentration. At larger distances from the coast, the offshore transport of nitrogen associated with the central subregion exceeds that originating from the southern

subregion because of a lower nutrient depletion caused by a more frequent subduction of upwelled water into deeper layers where phytoplankton growth is light-limited. The analysis of nitrogen allocation along trajectories reveals that the offshore transport of organic nitrogen is generally less than half of that of inorganic nitrogen and is mostly limited to the near surface in all subregions.





The pattern of offshore transport is characterized by the presence of latitudinally alternating offshore-onshore corridors indicating a strong contribution of mesoscale eddies and filaments to the mean transport. Major capes along the CanCS that are favorable to the formation of persistent filaments are associated with an enhanced offshore export of water and nitrogen. This results primarily from an enhancement of local export of water and nitrogen remotely upwelled (by up to a factor 4 to 9 for

C. Blanc) relative to non-cape areas. The offshore transport of nitrogen by remote upwelling exported by all capes constitutes more than $\frac{1}{3}$ of the total offshore transport of upwelling waters at all distances from the coast. All capes source the majority of water and nitrogen they export from remote upwelling. In the southern CanCS, most upwelled water and nitrogen traveling beyond 200 km from the coast are transported further offshore with no recirculation back to the coast. In contrast, the majority of water particles that upwell in the central (up to 65%) and northern (up to 80%) CanCS transported offshore recirculate

back to closer offshore distances in the area between 100 and 500 km offshore. This reflects the larger role played by recurrent mesoscale eddies in these regions. This enhanced recirculation results in an increase of the cross-shore transit times and a reduction in the efficiency of the offshore transport.

Finally, we found the supply of nitrogen by the Canary upwelling to the NATR and the NASE provinces, to amount to 43.7 mmol N m$^{-2}$yr$^{-1}$ and 7.3 mmol N m$^{-2}$yr$^{-1}$, respectively. This represents 45±15% and 3.5±1.5% of the total potential new

production in the two provinces, respectively The contribution of the Canary upwelling appears significant in comparison to other major sources of new nitrogen to the open North Atlantic Ocean. This emphasizes the importance of the CanCS upwelling in particular and EBUS in general as a key source of nutrient to the open ocean and the offshore transport as a mechanism of supply of these nutrients to the adjacent oligotrophic gyres, stressing the need for improving their representation in global coarse resolution models.

*Acknowledgements.*  Support for this research has come from the Center for Prototype Climate Modeling (CPCM), the New York University Abu Dhabi (NYUAD) Research Institute. Computations were performed at the High Performance cluster (HPC) of NYUAD, Dalma. NG acknowledges support by the Swiss National Science Foundation through the CALNEX project (Grant 149384) and by the Swiss Federal Institute of Technology Zurich (ETH Zurich). The authors are grateful to B. Blanke and N. Grima for making their ARIANE code available and to the NYUAD HPC team for technical support. The authors declare that they have no competing financial interests. The model code can

be accessed online (http://www.romsagrif.org).



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
