# Peer review of "A Lagrangian study of the contribution of the Canary coastal upwelling to the open North Atlantic nitrogen budget"

_Biogeosciences, 2020_

## Referee Comment (RC1) · Anonymous Referee #1 · 4 Mar 2020

The paper by Hailegeorgis et al. examined upwelling and transport patterns in the Canary Current System using a Lagrangian modeling approach. The authors identified latitudinal variability in water and nitrogen export, determined transit time of water parcels from the coast to the oligotrophic oceanic region, examined the role of major capes as drivers of upwelling and offshore transport, and quantified the coastal upwelling contribution to the nitrogen stock in the North Atlantic Tropical Gyral and the North Atlantic Subtropical Gyral East. Overall, I think that the paper does an interesting contribution to the understanding of nitrogen export patterns in the Canary Current System. However, important changes need to be made in order to recommend publication.

[Figure]

My main concerns are two:

1) The paper is not easy reading:

There are a lot of results and figures with multiple panels and supplement figures, which made fuzzy the paper's main points. There is some redundancy in the reported results. Subregional patterns not always showed important differences, so not sure if you always need reporting all subregional results in the paper main body (you can move some Figure's panels to the Supplement).

The manuscript length could be reduced, trying a better integration of the paper results. Results and discussion were mixed, which did not help to get the present study contribution. I am not an English native speaker, but below (specific comments) I made a series of suggestions to help making sentences more concise and clear.

The authors need explaining much clearly how all the Lagrangian patterns were calculated in the Method section.

2) Model validation:

The authors need to include vertical sections and vertical profiles from the modeled variables and compare them with observations (WOA sections would be OK). I would like to see whether the vertical patterns in the model outputs have any significant bias. Is the nutricline depth consistent with observed patterns across the model domain? Is the model reproducing well the seasonal variability in vertical patterns? It is not enough the correlation analysis in Fig. 1 from the Supplement, since well-correlated variables can have important differences in terms of magnitude.

Since a correlation analysis per se does not show a potential bias in the simulated variables, besides the annual mean patterns shown in Fig. 2. I would like to see a comparison for mean seasonal patterns of SST and surface chlorophyll.

To reduce paper length, I recommend including the model validation as an independent section in the Supplement.

Specific comments:

Please, refer to the supplement figures as Figure S1, Figure S2, etc. It was confusing when a paper and supplement figures were mentioned at the same time. As example, instead of using (Fig. 7 and Fig. 8, SI), use (Fig. 7 and Fig. S8).

Abstract

Pag.1, L11: "Our model analysis suggests that the vast majority of the upwelled waters originate from offshore and below the euphotic zone (70m depth), and once upwelled remain in the top 100m". I understand what you mean, but the statement is not clear. Consider that you defined upwelling as a water parcel crossing the 70 m depth level.

Introduction

Pag.2, L13-14: The sentence "leading to substantial modifications of the biogeochemical cycles there" is ambiguous. You could delete it.

Pag.2, L20: delete "potentially"

Pag.2, L32: delete "potentially"

Pag.3, L10: "surface jet associated with the upwelling flowing equatorward" => "surface jet associated with the coastal upwelling front, which flows equatorward"

Pag.4, L6: "and therefore did not estimate" => "so did not estimate"

Pag.4, L8-10: These sentences need additional work. Explain better but concise.

Pag.4, L13: What do you mean with "quantify the reach"

Pag.4, L13: "the spatial structure and the dominant timescales"

Pag.4, L23-24: and quantifying the offshore export

Pag.4, L24: "We also investigate"

Methods

Pag.5, L2: Was bottom remineralization included in the model? This is a relevant source of nitrogen, which can largely influence inorganic nitrogen patterns on the shelf. If it was not considered, you should mention it as another limitation in the study.

Pag.5, L14-15: Did you use a monthly climatology for wind stress? If this is the case, vertical mixing was probably underestimated. Did you compare the simulated mixed layer depth with observations?

Pag.5, L17-18: It was mentioned that the model was spun up by 12 years, and the study was based in simulation years 10 to 12. So does it mean that you used the last 3 years of your model spin up for the analysis? If this is the case, you have to report that the model was spun up by 9 years.

Figures 1 and 2: I suggest including the 200 m isobath (shelf break) as a contour line on the maps.

Pag.9, L2: It is Figure 2 not 1.

Pag.9, L14: Indicate that vertical mixing is not considered in the Lagrangian analysis

Pag.9, L25: Since the dominant circulation pattern in the CanCS appears to be along-shore, I am wondering why only oceanic particles were considered in the Lagrangian experiment. I would expect that upwelled particles also come from the northern and southern boundaries. Could the poleward undercurrent be a source for upwelled waters?

Results

Pag.13 L7: "onshore-offshore contrast" => "cross-shore differences"

Pag.13 L9: "followed by a second one at around 50 km from the shore" => "and a secondary maximum around 50 km"

Pag.13, L20. Shallow regions should also show upwelling because a distance of 50 km from the shelf break was considered for the analysis. Right? Clarify.

Pag.13, L25: what is the range considered for the upwelling zonal integration in Figure 6a (same for Figure 5a).

Pag.13, L26-28: Wondering if the shallow nutricline is linked to the high-nutrient SACW or not.

Pag13, L28-29: "The central subregion has a moderate nitrogen flux associated with upwelling while the northern subregion has the weakest upwelling flux of nitrogen" => "The weakest upwelling-driven nitrogen flux was in the northern subregion"

Pag.13, L30: "disproportionately lower" => "much lower"

Pag.15: It is important to note that a water parcel released in region A can be upwelled in subregion B or C. Beside, a water parcel upwelled in region A can be transported away to region B or C. Consequently, the parcel transport not necessarily depends on the oceanographic conditions in region A. If I'm right, I suggest revisiting all sentences describing subregional offshore transport results. As example, I would modify "the offshore transport is fastest in the central subregion and slowest in the northern subregion" by "the particle upwelled in the central and northern subregions displayed the fastest and slowest offshore transport, respectively". Besides, I would change "At larger distances from the coast (beyond 400 km), the offshore transport becomes faster in the southern subregion" by "The fastest water parcels beyond 400 km from the coast are those upwelled in the southern subregion"

Pag.15, L3: define "transit time".

Pag.15, L15-16: Why is this maximum at 150 km? Does it mean that a greater fraction of water parcels remains around this distance? Is this linked to patterns in alongshore circulation?

Pag.15: L22: "This is because of its low offshore transport efficiency and the relatively low volume of upwelling" => ''Reduced coastal upwelling and low offshore transport efficiency explain this pattern."

Pag.15, L22: Define transport efficiency.

Table 2: It may be informative reporting the standard deviation for transit time.

Pag.16, L3-5: "At close to 150 km from the coast (corresponding to the edge of the coastal upwelling stripe), the offshore transport of nitrogen reaches its maximum, reaching values as large as 500 Gmol yr$-1$. Thereafter, the offshore transport decreases exponentially" => "Around 150 km from the coast, the nitrogen transport reaches values as large as 500 Gmol yr-1 at 200 km, decreasing exponentially further offshore"

Pag.16, L7: "In spite of having the lowest offshore transport of water, the southern subregion exports the highest amount of nitrogen offshore at 200 km" => "Although offshore water transport was minima in the southern CanCS, this subregion has the greatest offshore export of nitrogen."

Pag.16, L9: "At larger distances from the coast, the situation reverses." => "This pattern reverses further offshore"

Pag.18, L2-4: I disagree with this statement: "so that beyond the nearshore 50 km region, inorganic nitrogen in the form of nitrate dominates the nitrogen pool at all distances from the coast". NO3 dominates almost everywhere, and its contribution to total nitrogen is actually much larger in the coastal region than in the oceanic region.

Pag.18, L21-34: It would be nice having a brief introductory explanation for the motivation of the analysis in this subsection. As example, evaluate the impact of subduction on transit time, impact of subduction on nutrient fluxes or nutrient cycling.

Pag.18, L21: "Upwelling particles that..." => "Upwelled particles that..."

Pag.18, L22: How do patterns in Fig. 8a,b from the supplement were calculated?

Pag.18, L24: I cannot see the secondary upwelling in the open ocean. Describe better.

Pag.18, L30: It is not evident for me why persistent filaments contribute to enhanced

subduction. Explain.

Figure 9: I recommend include this figure in the Supplement.

Pag.20. Indicate what the positive/negative values in Fig. 10 represent

Pag.21. Indicate what the positive/negative values in Fig. 11 represent

Pag.22, L26: "offshore transport of upwelled particles"

Pag.22, L30: I am not sure whether remotely upwelled water is a good term. I would prefer describing the results in terms of local and non-local upwelling.

Pag. 22, L32: "water upwelled"

Pag. 23: L1: "Corresponding enhancement in local nitrogen upwelling and export is seen only in" => "Increased offshore transport of nitrogen due to increased local upwelling is seen only in"

Caption of Figure 12: "Transport by water upwelling locally" => "Transport associated with locally upwelled water"

"Transport by particles that leave the coastal upwelling region at each cape or non-cape area but upwell remotely" => "Transport associated with remotely upwelled water"

Pag.24, L1-2: "The offshore transport of nitrogen by remote upwelling exported by all capes constitutes more than 30% of the total offshore transport of upwelling" => "Remotely upwelled waters that are transported offshore around major capes represent more than 30% of the total transport"

Pag.24, L4-5: "In fact, all capes source the majority of water and nitrogen they export from remote upwelling (Table 3). All capes also source more of their export from remote upwelling compared to the rest of the coast." => "Indeed, most of the water and nitrogen exported offshore around major capes is non-locally upwelled (Table 3)"

Pag.27, L10: For consistency use CanCS

Pag.27, L21-29: I did not understand. Please, explain better how did you estimate the CanCS contribution to the NATR and NASE provinces.

---

## Referee Comment (RC2) · Anonymous Referee #2 · 5 Mar 2020

General comments

This study aims to determine the contribution of nitrogen upwelled within the coastal region of the Canary Upwelling System to the nitrogen budget of the open ocean through a Lagrangian study relying on model outputs generated by a coupled physical-biogeochemical experiment. Authors also aim at describing the timescales, the reach and the structure of this offshore transport to quantify the role played by upwelling on nitrogen enrichment of the NATR and NASE provinces as defined by Longhurst. I'm not sure the study makes a significant contribution to the issue of nitrogen irrigation of the NATR and NASE provinces. There are several reasons for this. First, the authors

justify the originality of their study by the use of a Lagrangian approach as opposed to Eulerian approach which has been used in Auger et al. (2016) or Lovecchio et al. (2017, 2018) which have been mentioned in the present study. The authors in particular advance the capacity of their Lagrangian approach to define more faithfully the contribution of the coastal upwelling region in terms of nitrogen supply to the subtropical gyre (l5-10, page 4). This statement seems relevant with regard to the volume transported from the coastal region to the open sea, but much less obvious with regard to the transport of nitrogen. Indeed, the amount of nitrogen carried by each particle to a given location is quantified as the product of the particle associated volume and the concentration of the tracer associated with the particle when it reaches that location (l4-6 page 11). To my understanding of the methodology, the nitrogen concentration at a particular location does not necessarily come from the coastal upwelling but can be supplied locally, can change its chemical form or have a different origin. In addition, authors indicate some limitations of the biogeochemical model (absence of colimitation, absence of nitrogen fixation; l30-34, p31) but omit the potential role of different communities of phytoplankton. Indeed, the model used only represents a single phytoplankton community, the representation of diatom organisms (comprising a siliceous skeleton and likely to contribute significantly to the export of organic matter) could influence the export in the model. In terms of export, it has also been shown that the alternation of phase of intensification and relaxation of the upwelling favorable winds is important for the dynamics of the upwelling systems (significant efflorescence generation and sedimentation). The use of climatological wind in this study is likely to play a role in the results because it does not represent these alternations. These aspects should be mentioned in the limitations of the study. Then, the conclusions of the study highlight the importance of the Capes in the generation of filaments which represent privileged export sites but the influence of topographic accident on the generation of filaments has already been studied theoretically (eg Meunier et al., 2010), through hydrodynamic simulations and observations for certain filaments of the Canary upwelling system. The quantification of the overall contribution of filaments and the extension of

the source waters supplying the main filaments of the system nevertheless provides interesting information, even if the three-dimensional dimension of upwelling is becoming more and more essential in the literature targeting these upwelling regions. The role of mesoscale activity on residence times and the kinetics of transport from the coastal zone to the open sea is also part of the presented results. Mesoscale activity in the transition zone has been widely studied in all eastern boundary upwelling systems and fairly exhaustively in the northern part of the Canary system, in particular from the ROMS model (Mason et al., 2011 & 2012 ; Troupin et al., 2012). In the southern part of the area studied, the underestimation of EKE (Figure 1), an activity also highlighted by the occurrence of eddies in this region (Schutte et al., 2016), is not mentioned and is likely to impact the results in this region. The literature on the region is also to be completed, in particular to take into account recent studies by German, Senegalese and French teams. This update particularly concerns the southern part of the system which would allow the authors to describe their results more precisely. Hydrological conditions off Mauritania are described in Klenz et al. (2018), the vortex activity is studied in Schütte et al. (2016), the understanding of the dynamics of the Mauritanian current was revisited by Kounta et al. (2018), and the functioning of the Senegalese upwelling by Ndoye et al. (2014, 2015, 2017) or Capet et al. (2017). These studies point in particular to the importance of the Mauritanian current (to which I prefer the name West Africa Boundary Current; Kounta et al., 2018) on the dynamics of upwelling.

Finally, questions remain as to how to assess the contribution of nitrogen from coastal waters to new production. Indeed, I did not understand the use of VGPM models to quantify primary production knowing the large differences that exist in satellite-based models of primary production in the region (Gomes-Letona et al., 2017).

The manuscript is however well written, well illustrated with clean and condensed figures, the methodology is well described, and the main messages are clearly presented.

As a summary, the manuscript is of good quality but I hardly consider the results as really moving our understanding forward. The methodology that uses Ariane as a

Lagrangian tool is supposed to make a difference in the description, quantification of nitrogen irrigation of NATR and NASE provinces but I'm not convinced that it solves the issues faced by an Eulerian approach.

Specific comments

L21-23, page 2: Reformulate the sentence "Especially low-latitude ..." which is hardly understandable.

L12, page 3: The current of Mauritania must be considered in the light of the work of Kounta et al. (2018).

L30, page 4: Did you use ROMS or CROCO oceanic modeling system?

In this section 2.1.1, indicate the shallowest depth used at the coast (hmin parameter).

L26-28, page 5: EKE in the southern part of the domain is underestimated, please tell it and justify it.

L30-31: A warm bias seems to occur in the south, maybe a map of SST differences would make biases straightforward for the reader.

Figure 1: Arrows on a) and b) are almost invisible.

Figure 2: Validation on annual field does not inform on the ability of the model to correctly simulate the upwelling occurring in the southern part of the domain at the winter-spring time of the year. I believe it would strengthen confidence on the simulation to add this component.

L6-7, page 11: Why not telling here why you chose 70 m depth as upwelling criteria rather than explaining the reason much later.

L14-15, page 13: the description of the upwelling does not fit with the dynamic of the upwelling in the southern part of the domain (Ndoye et al., 2014, 2015, 2017; Capet et al., 2017)

L4, page 18: rather 300 km than 200 ?

Section 6: NPP and regenerated production are calculated by the coupled model. Why authors use satellite-based models here?

L31-34, page 31: Authors indicate some limitations of the biogeochemical model (absence of colimitation, absence of nitrogen fixation; l30-34, p31) but omit the potential role of different communities of phytoplankton. Indeed, the model used only represents a single community of phytoplankton, the representation of diatom type organisms (comprising a siliceous skeleton and likely to contribute significantly to the export of organic matter) could influence the export in the model. In terms of export, it has also been shown that the alternation of phases of intensification and relaxation of the upwelling favorable winds is important for the dynamics of the upwelling systems (blooms and sedimentation). The use of a climatological wind in this study is likely to play a role in the results because they don't represent these alternations. Affirmation lines 33-34 is true on an annual basis but could be false during the monsoon season when subtropical warm depleted open ocean waters invade the shelf in the southern part of the Canary Upwelling System.

L16-20, page 32: I agree that the western section is much more extended than the northern and southern exits but the role played by the West Africa Boundary Current (Mauritania Current here) plays a quite important role in cross-shore exchanges, it should be taken into account.

L16-19, page 33: The final conclusion states that this study emphasizes the need for improving the resolution of eastern boundary currents in global coarse resolution models. I think it would be fair to cite at least Large and Danabasoglu (2006) who stressed this point 15 years ago.

---

## Author Comment (AC1) · 1 Jun 2020

The authors are grateful for the thoughtful comments given by the two anonymous referees on our paper. Below, we respond to each point raised by the reviewers and explain the changes we've made to the manuscript accordingly. The reviewers' comments are shown below in italics writing while our response is indicated in a red font.

**Anonymous Referee #1**

We are grateful for the first referee's efforts in reviewing the manuscript. We believe our paper has significantly improved as a result of his/her comments. Below, we highlight our responses to the general and specific comments and explain the revisions we've made to the paper accordingly.

General comments

*The paper by Hailegeorgis et al. examined upwelling and transport patterns in the Canary Current System using a Lagrangian modeling approach. The authors identified latitudinal variability in water and nitrogen export, determined transit time of water parcels from the coast to the oligotrophic oceanic region, examined the role of major capes as drivers of upwelling and offshore transport, and quantified the coastal upwelling contribution to the nitrogen stock in the North Atlantic Tropical Gyral and the North Atlantic Subtropical Gyral East. Overall, I think that the paper does an interesting contribution to the understanding of nitrogen export patterns in the Canary Current System. However, important changes need to be made in order to recommend publication.*

*My main concerns are two:*

*1) The paper is not easy reading:*

*There are a lot of results and figures with multiple panels and supplement figures, which made fuzzy the paper's main points. There is some redundancy in the reported results. Subregional patterns not always showed important differences, so not sure if you always need reporting all subregional results in the paper main body (you can move some Figure's panels to the Supplement).*

In order to solve the redundancy and lengthiness issues in the main paper, we will merge some subsections as well as remove figures/panels (some of which will be moved to the supplementary). More specifically:

- We will move the model validation (including Fig1, Fig 2 and Fig3) into the Supplementary section following the reviewer comment 6.
- We will merge subsections 3.1 and 3.2 so there is now no subsection in Section 3.
- We will merge Figure 5 and Figure 6 in one single figure and move some of their sub-panels to the supplementary.

- We will merge and consolidate subsections 4.4 and 4.5 into one subsection 4.4 titled "Structure of offshore transport", which includes results on depth structure and horizontal structure of the offshore transport.
- Figure 9 will be moved to the supplementary as suggested by the reviewer (see below)
- Figure 10 panels c-h will be moved to the supplementary
- We will merge Figure 10 (panels a-b) and Fig 11 in one figure.

*2) The manuscript length could be reduced, trying a better integration of the paper results. Results and discussion were mixed, which did not help to get the present study contribution.*

We will significantly shorten the paper and consolidate multiple sections (please see our response to previous comment).

*I am not an English native speaker, but below (specific comments) I made a series of suggestions to help making sentences more concise and clear.*

*3) The authors need explaining much clearly how all the Lagrangian patterns were calculated in the Method section.*

We will move some important details of the Lagrangian experiment from the supplementary text into Section 2.2.

We will also improve the description of the Lagrangian experiment by adding a schematic that summarizes the different steps followed to quantify the Lagrangian offshore transport (shown below).

| Step 1 | Step 2 | Step 3 |
|---|---|---|
| Transport driven particle release (ARIANE Quantitative Experiment) | Particle tracking (ARIANE Qualitative Experiment) | Upwelling identification |
| The particle release strip is set 50km westward of the 200m bathymetry isobath to be the western boundary of the coastal upwelling domain | Particle trajectories are initialized at the initial conditions in the quantitative experiment (Step 1) | All particles that cross 70m depth (the base of the euphotic zone) upwards while still in the coastal domain (before exiting the coast) are considered **upwelled particles** |
| Particles are released at each vertical cell along the particle release strip on each day of one year of the experiment according to the coastward transport, where each particle is associated with 0.01 Sv of transport or less | Trajectories are tracked for 720 days or until a particle leaves the experiment domain (whichever happens first) -- we call this **exiting the domain**. This includes the whole experiment domain as opposed to the coastal upwelling area in the quantitative experiment. | For all upwelled particles, their trajectories (obtained in Step 2) are considered from their upwelling until they exit the domain |
| Particle trajectories are terminated at 720 days or when they exit the coastal upwelling area (whichever comes first) -- we call this **exiting the coast**. This domain is limited to the area between the particle release strip and the West African coast. | Trajectory information (incl. position, velocity, nearby tracer concentration) is saved on a 1-day interval | In order to account for an upwelled water mass that is double-counted (see Steps 1 & 2), if an upwelled particle leaves the coastal domain but later re-enters and re-upwells, its trajectory since its second upwelling will be discarded (the particle's time of exiting the domain is shortened). This is because the subsequent trajectory will have been considered as an upwelling of a separate particle released at the water mass's re-entry into the coastal domain. |
| For each particle, its associated volume, position, time, velocity and nearby tracer concentration are saved at its entry into the coastal domain and on its exit from the coast | In the case of a double-counted water mass (see Step 1), it is represented by two particle trajectories starting from its re-entry into the coastal domain | |
| Note that there may be **double-counting** of a water mass that occurs if a water mass enters the coastal domain (initiating release of one particle), exits the coast and re-enters (initiating release of a second particle) | | |

We then will add details on ARIANE's quantitative and qualitative experiments from what used to be in the supplementary text. As part of the quantitative experiment, we explain how we save the initial condition (including associated volume) of a particle release and keep track of particle recirculation in and out of the coastal upwelling area. Many important details are mentioned here including the particle release period of one year, the release of particles over all the cells vertically along the particle release strip, the maximum volume associated with particles, etc. As part of the qualitative experiment, we explain details including the maximum length of trajectories and the frequency at which they are saved, the information saved about each particle along its trajectory and the portion of particles that leave the experiment domain early.

We also indicate in this subsection that (1) ARIANE neglects vertical mixing in its computation of particle trajectories and (2) although the validity of the assumption of incompressibility of ocean water conserves the water flow in and out of streamtubes, ARIANE doesn't have a stochastic noise term to model diffusion along particle trajectories, although diffusion is included in the model.

*4) Model validation:*

*The authors need to include vertical sections and vertical profiles from the modeled variables and compare them with observations (WOA sections would be OK). I would like to see whether the vertical patterns in the model outputs have any significant bias. Is the nutricline depth consistent with observed patterns across the model domain? Is the model reproducing well the seasonal variability in vertical patterns? It is not enough the correlation analysis in Fig. 1 from the Supplement, since well-correlated variables can have important differences in terms of magnitude.*

We agree with the reviewer. We have produced validations of vertical cross sections of temperature, salinity and nitrate (WOA) for each subregion and for each season.

The model generally does a good job in reproducing the vertical variability in observations for all three variables across the different seasons. We present our evaluation with more detail under Model Evaluation which will be presented in the supplementary material.

As examples, the figures below show seasonal validations of vertical cross sections of (from left to right) temperature at 16N, salinity at 25N and nitrate at 30N. Each of these sample plots is taken from a distinct subregion in our experiment.

[Figure]

5) *Since a correlation analysis per se does not show a potential bias in the simulated variables, besides the annual mean patterns shown in Fig. 2. I would like to see a comparison for mean seasonal patterns of SST and surface chlorophyll.*

We have produced seasonal validation maps of SST (Pathfinder) and chlorophyll (SeaWIFS) (shown below, in that order)

[Figure]

Furthermore, we have added a comparison of modeled mixed layer depth (MLD) to observations in all seasons (please see our response to comment #20)

*6) To reduce paper length, I recommend including the model validation as an independent section in the Supplement.*

Following the reviewer's suggestion, we will move the model validation to the supplementary.

Specific comments:

*7) Please, refer to the supplement figures as Figure S1, Figure S2, etc. It was confusing when a paper and supplement figures were mentioned at the same time. As example,*

*instead of using (Fig. 7 and Fig. 8, SI), use (Fig. 7 and Fig. S8).*

We will make these changes.

*8) Abstract*
*Pag.1, L11: "Our model analysis suggests that the vast majority of the upwelled waters*
*originate from offshore and below the euphotic zone (70m depth), and once upwelled*
*remain in the top 100m". I understand what you mean, but the statement is not clear.*
*Consider that you defined upwelling as a water parcel crossing the 70 m depth level.*

We agree with the referee. We will change this statement to:
"Our model analysis suggests that the vast majority of the upwelled waters remain in the top
100m."

*9) Introduction*
*Pag.2, L13-14: The sentence "leading to substantial modifications of the biogeochemical cycles*
*there" is ambiguous. You could delete it.*

The sentence will be deleted.

*10) Pag.2, L20: delete "potentially"*

The correction will be made.

*11) Pag.2, L32: delete "potentially"*

The correction will be made.

*12) Pag.3, L10: "surface jet associated with the upwelling flowing equatorward" => "surface jet*
*associated with the coastal upwelling front, which flows equatorward"*

The correction will be made.

*13) Pag.4, L6: "and therefore did not estimate" => "so did not estimate".*

The correction will be made.

*14) Pag.4, L8-10: These sentences need additional work. Explain better but concise.*

When it comes to the quantification of the contribution of the Canary upwelling to the open ocean
nitrogen budget, the Lagrangian approach presents a couple of advantages relative to the Eulerian
approach. Because of its focus on water particle trajectories, Lagrangian tracking of water masses

is better suited for the analysis of connectivity between the coastal and the open ocean regions. Furthermore, the Lagrangian method can be used to derive conditional statistics where subsets of particles that fulfill certain criteria are analyzed. This is useful for instance to restrict the analysis of offshore transport to upwelling particles only.

This will be better explained in the revised manuscript.

*15) Pag.4, L13: What do you mean with "quantify the reach"*

We will modify this to 'quantify the offshore reach'.

*16) Pag.4, L13: "the spatial structure and the dominant timescales"*

We will make this change.

*17) Pag.4, L23-24: and quantifying the offshore export*

We will make this change.

*18) Pag.4, L24: "We also investigate"*

We will make this change.

*19) Methods*
*Pag.5, L2: Was bottom remineralization included in the model? This is a relevant source of nitrogen, which can largely influence inorganic nitrogen patterns on the shelf. If it was not considered, you should mention it as another limitation in the study.*

Yes, the sinking particulate organic matter that reaches the seafloor is remineralized back into ammonium at a slower rate (0.003 d−1 ) than in the water column (0.03 d−1 for small detritus and 0.01 d−1 for large detritus). We will add that explanation and a reference to Gruber et al. (2006).

*20) Pag.5, L14-15: Did you use a monthly climatology for wind stress? If this is the case, vertical mixing was probably underestimated. Did you compare the simulated mixed layer depth with observations?*

We used a monthly wind climatology (QuickSCAT SCOW). We have added a comparison of the modeled mixed layer depth (MLD) with observations (will be shown in supplementary materials) for all seasons. Using a monthly climatology forcing likely contributes to the underestimation of EKE shown in Fig 1. However, much of the spatial and seasonal variability in upper ocean mixing is still captured by the model as can be seen in the figure below. This is particularly true in the summer

and fall when the simulated MLD agrees the most with observations. This will be explicitly discussed in the revised manuscript.

[Figure]

*21) Pag.5, L17-18: It was mentioned that the model was spun up by 12 years, and the study was based in simulation years 10 to 12. So does it mean that you used the last 3 years of your model spin up for the analysis? If this is the case, you have to report that the model was spun up by 9 years.*

Yes, we did spin up for 9 years then used years 10-12 to run the experiment. This correction will be made.

*22) Figures 1 and 2: I suggest including the 200 m isobath (shelf break) as a contour line on the maps.*

This change has been made as can be seen in the reproduction of Figure 2 below.

[Figure]

*23) Pag.9, L2: It is Figure 2 not 1.*

We will make this change.

*24) Pag.9, L14: Indicate that vertical mixing is not considered in the Lagrangian analysis*

This will be clarified.

*25) Pag.9, L25: Since the dominant circulation pattern in the CanCS appears to be along-shore, I am wondering why only oceanic particles were considered in the Lagrangian experiment. I would expect that upwelled particles also come from the northern and southern boundaries. Could the poleward undercurrent be a source for upwelled waters?*

It is true that potential upwelling driven by waters entering the coastal zone at 14N and 35N were not included. We acknowledge this among the study's caveats.

Our choice to only allow open ocean particle releases was dictated by technical constraints in Ariane that prevent individual segments to be considered simultaneously as entry and exit sections. Yet, in order to quantify the potential error associated with our simplification we have run a separate experiment where particles were allowed to enter the coastal ocean from the southern and northern boundaries of the upwelling strip. Our analysis reveals that we are missing only a very small amount of water by our choice, i.e., only about 1% and 3% of the total volume tracked in this study come through the northern and southern boundaries , respectively.

Furthermore, given the high rate of recirculation near the coast, a vast proportion of the particles that may have entered the coast alongshore, once upwelled and exported offshore, are likely to return to the coast from the open ocean, in which case they would be sampled in our particle release.

We conclude that discarding particles entering the coastal ocean from the southern and northern boundaries of the upwelling strip is likely to cause only a limited error in our quantification of the offshore transport.

We will further highlight this caveat and discuss its potential implications in the revised manuscript based on the arguments developed above.

*26) Results*
*Pag.13 L7: "onshore-offshore contrast" => "cross-shore differences"*

We will make this change.

*27) Pag.13 L9: "followed by a second one at around 50 km from the shore" => "and a secondary maximum around 50 km"*

We will make this change.

*28) Pag.13, L20. Shallow regions should also show upwelling because a distance of 50 km from the shelf break was considered for the analysis. Right? Clarify.*

The reason there is no upwelling at the very coast in these regions is that the bathymetry is shallower than 70m, therefore making it impossible to meet our criteria for upwelling where a

particle crosses the 70m depth. The minimum bathymetry allowed by the model is 50 m and that will be added to the model's description in section 2.1.1 as Reviewer 2 has also requested (see below). However, as mentioned by the referee we sample upwelling further offshore in these locations. Therefore, for more clarity we will change the statement to: "our experiment identifies limited coastal upwelling south of Cape Blanc and between Capes Barbas and Bojador because their bathymetry is shallower than the 70 m upwelling depth criterion used here".

*29) Pag.13, L25: what is the range considered for the upwelling zonal integration in Figure 6a (same for Figure 5a).*

We integrate zonally across the coastal stripe (Between the coast and  50 km westward of the 200m bathymetry). We will add this clarification to the captions of Fig5 and Fig6.

*30) Pag.13, L26-28: Wondering if the shallow nutricline is linked to the high-nutrient SACW or not.*

No. The shallow nutricline is a consequence of the shallow thermocline which is caused by the stronger stratification in the tropics in comparison with the high latitudes.

*31) Pag13, L28-29: "The central subregion has a moderate nitrogen flux associated with upwelling while the northern subregion has the weakest upwelling flux of nitrogen" => "The weakest upwelling-driven nitrogen flux was in the northern subregion"*

We will make this change.

*32) Pag.13, L30: "disproportionately lower" => "much lower"*

We prefer to use the statement  "disproportionately lower" to better highlight the fact that the ratio of upwelled nitrogen to upwelled water volume is higher near the coast than it is further offshore as suggested by Figures 5 & 6.

*33) Pag.15: It is important to note that a water parcel released in region A can be upwelled in subregion B or C. Beside, a water parcel upwelled in region A can be transported away to region B or C. Consequently, the parcel transport not necessarily depends on the oceanographic conditions in region A. If I'm right, I suggest revisiting all sentences describing subregional offshore transport results. As example, I would modify "the offshore transport is fastest in the central subregion and slowest in the northern subregion" by "the particle upwelled in the central and northern subregions displayed the fastest and slowest offshore transport, respectively". Besides, I would change "At larger distances from the coast (beyond 400 km), the offshore transport becomes faster in the southern subregion" by "The fastest water parcels beyond 400 km from the coast are those upwelled in the southern subregion"*

We will follow this suggestion and correct the way we express transport from each subregion accordingly.

*34) Pag.15, L3: define "transit time".*

We will add the transit time definition from the caption of Table 2 here.

*35) Pag.15, L15-16: Why is this maximum at 150 km? Does it mean that a greater fraction of water parcels remains around this distance? Is this linked to patterns in alongshore circulation?*

In the revised manuscript we explain that the distance of maximum transport corresponds to the distance reached by most sampled upwelled water (nitrogen). It is around 150km (that roughly corresponds to the width of the upwelling stripe) because at closer distances to the coast (<100-150km), upwelled volume is only partially sampled. Farther distances (>100-150km) are never reached by a proportion of particles because of recirculation retaining them close to the coast and alongshore transport exporting some particles out of the model domain.

*36) Pag.15: L22: "This is because of its low offshore transport efficiency and the relatively low volume of upwelling" => ''Reduced coastal upwelling and low offshore transport efficiency explain this pattern."*

We will make this change.

*37) Pag.15, L22: Define transport efficiency.*

Transport efficiency (the ratio of net offshore transport at a given distance to upwelling volume or nitrogen amount) will be defined in the Lagrangian experiment details (Section 2.2).

*38) Table 2: It may be informative reporting the standard deviation for transit time.*

We will replace Table 2 with a graph of transit times featuring error bars.

*39) Pag.16, L3-5: "At close to 150 km from the coast (corresponding to the edge of the coastal upwelling stripe), the offshore transport of nitrogen reaches its maximum, reaching values as large as 500 Gmol yr−1. Thereafter, the offshore transport decreases exponentially" => "Around 150 km from the coast, the nitrogen transport reaches values as large as 500 Gmol yr-1 at 200 km, decreasing exponentially further offshore"*

We will make this change.

*40) Pag.16, L7: "In spite of having the lowest offshore transport of water, the southern subregion exports the highest amount of nitrogen offshore at 200 km" => "Although offshore water transport was minima in the southern CanCS, this subregion has the greatest offshore export of nitrogen."*

We will make this change.

*41) Pag.16, L9: "At larger distances from the coast, the situation reverses." => "This pattern reverses further offshore"*

We will make this change.

*42) Pag.18, L2-4: I disagree with this statement: "so that beyond the nearshore 50 km region, inorganic nitrogen in the form of nitrate dominates the nitrogen pool at all distances from the coast". NO3 dominates almost everywhere, and its contribution to total nitrogen is actually much larger in the coastal region than in the oceanic region.*

We agree with the referee that $NO_3^-$ dominates nearly everywhere. Yet, in the nearshore 50km region, the sum of all organic nitrogen (in both plankton and detritus forms) can exceed that of nitrate as shown by the fraction of organic N in the offshore transport (red dashed curve) in Figure 8.

*43) Pag.18, L21-34: It would be nice having a brief introductory explanation for the motivation of the analysis in this subsection. As example, evaluate the impact of subduction on transit time, impact of subduction on nutrient fluxes or nutrient cycling.*

We will add introductory statements in the beginning of this subsection on the potential impact of vertical structure of offshore transport (subduction) on transit times and offshore transport efficiency to serve as motivation for the depth structure analysis. At the end of the subsection, these potential links are revisited based on the analysis that has just been presented. In general, subduction increases transit times (slows down offshore transport) since velocities deeper in the water column are smaller. However, subduction can also increase the efficiency of offshore transport by minimizing the depletion of nitrogen in surface waters.

Elsewhere in the original draft, we've also related subduction of water particles with filament activity (same subsection on page 18) and with nitrogen offshore transport efficiency (Pg 31, Sec 8, Par 3 & Pg 32, Sec 9, Par 1).

*44) Pag.18, L21: "Upwelling particles that..." => "Upwelled particles that..."*

We will make this change.

*45) Pag.18, L22: How do patterns in Fig. 8a,b from the supplement were calculated?*

To make Figure 8 a (or b), we first considered all downward (or upward) movements of all particles across the horizontal plane at 50m depth. After finely binning the horizontal surface across the model, we added the associated volumes of all particles that crossed this depth downward (or

upward) for each binned area throughout the 2 year particle trajectory experiment. We will add more detail to the caption of Fig. 8S to show this more clearly.

Although Fig. 8c gives the dominant vertical transport (upward or downward) in the modeled region, panels (a) and (b) also complement this information by giving the full picture of where the upward and downward transport occurs during the whole 2 year experiment. Panels (a) and (b), for example, show that although the coast is dominated by an upwelling flux, it also features significant downwelling.

*46) Pag.18, L24: I cannot see the secondary upwelling in the open ocean. Describe better.*

It is true that Figure 10d doesn't show significant upwelling flux for particles upwelled in the southern subregion except for a subsurface upward transport maxima in the first 300km from coast. So we will only cite Figure 8S(c), which does show strong net upward transport (upwelling) in the open ocean in the southern CanCS, which has significant overlap with the trajectory of the upwelling from this subregion.

*47) Pag.18, L30: It is not evident for me why persistent filaments contribute to enhanced subduction. Explain.*

Here, we are referring to the role of filaments in squirting cold upwelled water offshore and the subsequent subduction of this cold water due to its high density compared to the open ocean's warmer surface water. We will add a sentence stating this fact including a reference to Lovecchio et al. (2018), who found filaments to cause subduction of organic nitrogen, for example, to depths larger than 100m.

*48) Figure 9: I recommend include this figure in the Supplement.*

We will make this change.

*49) Pag.20. Indicate what the positive/negative values in Fig. 10 represent*

We will make this change.

*50) Pag.21. Indicate what the positive/negative values in Fig. 11 represent*

We will make this change.

*51) Pag.22, L26: "offshore transport of upwelled particles"*

We will make this change.

*52) Pag.22, L30: I am not sure whether remotely upwelled water is a good term. I would prefer describing the results in terms of local and non-local upwelling.*

We will change all references to remote upwelling to non-local upwelling.

*53) Pag. 22, L32: "water upwelled"*

We will make this change.

*54) Pag. 23: L1: "Corresponding enhancement in local nitrogen upwelling and export is seen only in" => "Increased offshore transport of nitrogen due to increased local upwelling is seen only in"*

We will make this change.

*55) Caption of Figure 12: "Transport by water upwelling locally" => "Transport associated with locally upwelled water"*
*"Transport by particles that leave the coastal upwelling region at each cape or non-cape area but upwell remotely" => "Transport associated with remotely upwelled water"*

We will make these changes.("Transport associated with remotely upwelled water at each cape or non-cape area").

*56) Pag.24, L1-2: "The offshore transport of nitrogen by remote upwelling exported by all capes constitutes more than 30% of the total offshore transport of upwelling" => "Remotely upwelled waters that are transported offshore around major capes represent more than 30% of the total transport"*

We will make this change.

*57) Pag.24, L4-5: "In fact, all capes source the majority of water and nitrogen they export from remote upwelling (Table 3). All capes also source more of their export from remote upwelling compared to the rest of the coast." => "Indeed, most of the water and nitrogen exported offshore around major capes is non-locally upwelled (Table 3)"*

The second sentence is meant to show the high export of non-local upwelling that occurs at capes compared to the non-cape coast. But the phrasing was unclear so we will modify it slightly.

*58) Pag.27, L10: For consistency use CanCS*

We will make this change.

*59) Pag.27, L21-29: I did not understand. Please, explain better how did you estimate the CanCS contribution to the NATR and NASE provinces.*

We have added a brief clarification of our calculation. Briefly, it is as follows.

We explain that whenever a particle enters a province within the top 100m, the nitrogen it carries into the province is added to the particle's source subregion's contribution to the province. On the contrary, when a particle from a given subregion leaves a province within the top 100m, the nitrogen it carries with it when it leaves the province is subtracted from the contribution of the subregion of the particle to the province.

We use precise Longhurst province boundaries for our analysis. Analysis of crossing into and out of a province is determined based on a daily resolution of particle positions since that is the resolution of our Lagrangian trajectories.

---

## Author Comment (AC2) · 1 Jun 2020

The authors are grateful for the thoughtful comments given by the two anonymous referees on our paper. Below, we respond to each point raised by the reviewers and explain the changes we've made to the manuscript accordingly. The reviewers' comments are shown below in italics writing while our response is indicated in a red font.

**Anonymous Referee #2**

We would like to thank the second reviewer for the comments that have, we hope, significantly improved our manuscript. Below, we highlight our responses, point by point, to the reviewer's general and specific comments and indicate the revisions we will make to the paper accordingly.

General comments

*This study aims to determine the contribution of nitrogen upwelled within the coastal region of the Canary Upwelling System to the nitrogen budget of the open ocean through a Lagrangian study relying on model outputs generated by a coupled physical-biogeochemical experiment. Authors also aim at describing the timescales, the reach and the structure of this offshore transport to quantify the role played by upwelling on nitrogen enrichment of the NATR and NASE provinces as defined by Longhurst. I'm not sure the study makes a significant contribution to the issue of nitrogen irrigation of the NATR and NASE provinces. There are several reasons for this.*

1) *First, the authors justify the originality of their study by the use of a Lagrangian approach as opposed to Eulerian approach which has been used in Auger et al. (2016) or Lovecchio et al. (2017, 2018) which have been mentioned in the present study. The authors in particular advance the capacity of their Lagrangian approach to define more faithfully the contribution of the coastal upwelling region in terms of nitrogen supply to the subtropical gyre (l5-10, page 4). This statement seems relevant with regard to the volume transported from the coastal region to the open sea, but much less obvious with regard to the transport of nitrogen. Indeed, the amount of nitrogen carried by each particle to a given location is quantified as the product of the particle associated volume and the concentration of the tracer associated with the particle when it reaches that location (l4-6 page 11). To my understanding of the methodology, the nitrogen concentration at a particular location does not necessarily come from the coastal upwelling but can be supplied locally, can change its chemical form or have a different origin.*

We do agree with the referee that in contrast to water volume, tracing the transport of nitrogen is somewhat more difficult given the chemical transformations between inorganic and organic nitrogen. In addition, subgrid mixing is not represented in our Lagrangian particle tracking but can affect nitrogen concentrations in ways that are not accounted for in our transport estimates.

Yet, as has been shown also by Frischknecht et al. (2018), the Lagrangian method permits a lot of new insight into the offshore transport of nitrogen, since total nitrogen, i.e., the sum of inorganic and organic nitrogen is conserved except for the part that is sinking, and that part that is being

supplied through mixing. Indeed, along the way, nitrogen can be incorporated into organic matter and then being recycled again, but if it is tracked by our algorithm, this nitrogen is still coming from the coastal upwelling.

The component we lose through sinking does not affect our conclusions, since this component is lost to the ocean interior, from where it will not find its way back into the waters that are transported offshore. More importantly is our lack of consideration of the vertical mixing. We have good evidence that this component is relatively small. First, the total amount of nitrogen is decreasing with offshore distance, and not increasing. In fact, the decrease is driven entirely by the sinking component, and the spatial distribution of this loss fits well the spatial distribution of the export of organic nitrogen (Figure 7). In particular, we see a decline in total nitrogen as a function of distance to the coast that is sharper than for water volume (Fig 7). And this decline is larger for water particles originating from the southern subregion that are transported at the shallowest depths (Fig 9). If the supply of nitrogen from surrounding waters to upwelling waters due to mixing were large enough to cancel the loss due to organic matter sinking, there would be no such a sharp decline in the offshore transport of nitrogen as a function of distance to coast. This suggests that although the potential changes in nitrogen due to subgrid mixing can locally be important, they are unlikely to affect the large-scale transport estimates in a significant way.

Yet, we acknowledge that the lack of a representation of mixing in Ariane is an important caveat that not only can affect particles' depth, but also potentially their nitrogen content and hence locally our offshore transport estimate. This will be stated explicitly in the discussion of the method caveats.

2) *In addition, authors indicate some limitations of the biogeochemical model (absence of colimitation, absence of nitrogen fixation; l30-34, p31) but omit the potential role of different communities of phytoplankton. Indeed, the model used only represents a single phytoplankton community, the representation of diatom organisms (comprising a siliceous skeleton and likely to contribute significantly to the export of organic matter) could influence the export in the model. In terms of export, it has also been shown that the alternation of phase of intensification and relaxation of the upwelling favorable winds is important for the dynamics of the upwelling systems (significant efflorescence generation and sedimentation). The use of climatological wind in this study is likely to play a role in the results because it does not represent these alternations. These aspects should be mentioned in the limitations of the study.*

We agree that similar to other state-of-the-art models, our model (especially the biogeochemical module) has other limitations beyond what we have already acknowledged in the original version of the manuscript. Yet, the fact that the simulated distributions of nitrate and its seasonality agree relatively well with the observations (as shown in the new validation figures; see also our response to comment 4 by reviewer 1) suggests that the impact of these limitations on the study conclusions is likely limited.

Nevertheless, following the referee's suggestions we have added two additional potential model limitations in the caveat section: 1) the fact that the model does not represent multiple phytoplankton groups and 2) the use of climatological winds lacking high-frequency variability that may lead to a misrepresentation of some aspects of the complex upwelling dynamics.

3) *Then, the conclusions of the study highlight the importance of the Capes in the generation of filaments which represent privileged export sites but the influence of topographic accident on the generation of filaments has already been studied theoretically (eg Meunier et al., 2010), through hydrodynamic simulations and observations for certain filaments of the Canary upwelling system. The quantification of the overall contribution of filaments and the extension of the source waters supplying the main filaments of the system nevertheless provides interesting information, even if the three-dimensional dimension of upwelling is becoming more and more essential in the literature targeting these upwelling regions.*

We agree with the referee in that previous studies like Meunier et al. (2010) and Troupin et al. (2012) have demonstrated the importance of coastal topography and capes in particular in the formation of coastal filaments. Therefore, we will add references to these previous works in the revised manuscript.

In particular:
- on page 22, Ln 14, we will change the statement :"Coastal filaments along the West African coast can occur everywhere, but it is well established that the majority of the filaments are persistently associated with the major capes along the coast" to "Coastal filaments can occur everywhere anywhere on the coast in the CanCS, but previous studies have shown that capes can facilitate their formation (Meunier et al., 2010; Troupin et al., 2012)".
- on page 24, Ln 10, we will cite Meunier et al. (2010) and Troupin et al. (2012) to highlight that the alongshore advection can interact with capes to result in the formation of a coastal filament that then exports upwelled water to the open ocean.

4) *The role of mesoscale activity on residence times and the kinetics of transport from the coastal zone to the open sea is also part of the presented results. Mesoscale activity in the transition zone has been widely studied in all eastern boundary upwelling systems and fairly exhaustively in the northern part of the Canary system, in particular from the ROMS model (Mason et al., 2011 & 2012 ; Troupin et al., 2012).*

We agree that previous studies mentioned by the referee have studied the mesoscale variability in the northern Canary system. We will add references to these papers in the revised manuscript.
In particular, we will cite:

- Mason et al. (2011) in section 5 among the papers we cite showing mesoscale variability and overall transport complexity in the Canary.

- Mason et al. (2012) will be cited in our literature review in the introduction section and section 5.2 on recirculation.

- Troupin et al. (2012) will be cited in our description of the Cape Ghir filament.

- Barton and Aristegui (2004) is an additional reference that we will add on mesoscale activity in the Canary (see subsection 5.2).

5) *In the southern part of the area studied, the underestimation of EKE (Figure 1), an activity also highlighted by the occurrence of eddies in this region (Schutte et al., 2016), is not mentioned and is likely to impact the results in this region. The literature on the region is also to be completed, in particular to take into account recent studies by German, Senegalese and French teams. This update particularly concerns the southern part of the system which would allow the authors to describe their results more precisely. Hydrological conditions off Mauritania are described in Klenz et al. (2018), the vortex activity is studied in Schütte et al. (2016), the understanding of the dynamics of the Mauritanian current was revisited by Kounta et al. (2018), and the functioning of the Senegalese upwelling by Ndoye et al. (2014, 2015, 2017) or Capet et al. (2017). These studies point in particular to the importance of the Mauritanian current (to which I prefer the name West Africa Boundary Current; Kounta et al., 2018) on the dynamics of upwelling.*

We agree with the referee that the model underestimates the EKE  in the coastal area of the southern subregion, and particularly so south of Cape Verde. Following the referee's comment, we will explicitly mention the underestimation of the EKE in that region and its potential implications among the study's caveats.

Following the referee's suggestion, we also have expanded our review of the literature in the region. In particular, we will cite the following works in the revised manuscript:

- Schutte et al. (2016) and Kounta et al. (2018): will be cited in subsection 5.2 (as well as in the Introduction section) to highlight the potential importance of the Mauritanian current and eddies in the region in fueling offshore export of water.
- We also cite Glessmer et al. (2009) and Peña-Izquierdo et al. (2015) to emphasize the importance of the Mauritanian current in the Introduction section.
- Klenz et al. (2018): will be cited in section 5.1 to highlight the importance of both the poleward (Mauritanian) current as well as the equatorward Northern Atlantic (Canary) current, particularly during winter, as a source of upwelling in the southern subregion. This helps explain the patterns in Figure 13 for the source waters of exports at capes in the southern subregion.
- Ndoye et al. (2017) and Capet et al. (2017): will be cited in our description of the Cape Verde filament and the importance of the local mesoscale activity in section 5.2.

6) *Finally, questions remain as to how to assess the contribution of nitrogen from coastal waters to new production. Indeed, I did not understand the use of VGPM models to quantify primary*

*production knowing the large differences that exist in satellite-based models of primary production in the region (Gomes-Letona et al., 2017).*

We used satellite-based (VGPM) NPP estimates because of their synoptic-scale coverage, which individual in-situ estimates lack. However, we are aware of the important uncertainties associated with this (and other satellite-based) product(s). Therefore, we will add productivity estimates based on in-situ measurements that are available for some Longhurst provinces (Tilstone et al., 2009) as well as estimates from the CbPM model (Westberry et al., 2008).

Indeed, in-situ estimate of primary production for the NATR based on Carbon-14 uptake from Tilstone et al. (2009) will be added. This estimate (1377mmol N m-2 yr-1) is less than that derived from satellite data (1753 mmol N m-2 yr-1). This suggests that the contribution of the Canary upwelling nitrogen supply can be locally even more important in relative terms, than what our initial estimates have implied. Using a carbon-based productivity model (CbPM), NPP estimates for the NATR and NASE both amount to around 2000 mmol N m-2 yr-1. This also suggests that the CanCS's contribution to the NASE is locally more important than what the province's VGPM value (2139 mmol N m-2 yr-1) suggested.

We will update Table 4 and the discussion of the potential contribution of the nitrogen transport to the total NP estimates in the NATR and NASE provinces accordingly.

Finally, Gomez-Letona (2017) compares three alternative estimates of PP and finds divergences between them. However, their estimates and comparisons are focused on the coastal area while we were primarily interested in finding estimates for the larger adjacent Longhurst provinces so that we could compare offshore transport of nutrients in our model with the provinces' total budgets.

7) *The manuscript is however well written, well illustrated with clean and condensed figures, the methodology is well described, and the main messages are clearly presented.*

We thank the referee for his/her positive and encouraging comment.

8) *As a summary, the manuscript is of good quality but I hardly consider the results as really moving our understanding forward. The methodology that uses Ariane as a Lagrangian tool is supposed to make a difference in the description, quantification of nitrogen irrigation of NATR and NASE provinces but I'm not convinced that it solves the issues faced by an Eulerian approach.*

Again we thank the referee for praising the quality of the manuscript. We believe that while it comes with important limitations, our study does improve the quantification of the coastal upwelling supply of nitrogen to the open ocean, relative to the Eulerian approach. We detail our reasoning in our response to the previous comment #1by the same reviewer.

Specific comments

9) *L21-23, page 2: Reformulate the sentence "Especially low-latitude ..." which is hardly understandable.*

We will make this change.

10) *L12, page 3: The current of Mauritania must be considered in the light of the work of Kounta et al. (2018).*

Done. See response to comment 25above.

11) *L30, page 4: Did you use ROMS or CROCO oceanic modeling system?*

We used ROMS-AGRIF version 3.1.1 (which shares the same code with the current version of CROCO).

12) *In this section 2.1.1, indicate the shallowest depth used at the coast (hmin parameter).*

We'll indicate the 50m lowest bathymetry in section 2.1.1 accordingly.

13) *L26-28, page 5: EKE in the southern part of the domain is underestimated, please tell it and justify it.*

We indicate in the revised manuscript that the model underestimates EKE in the southernmost part of the CanCS region as well.

14) *L30-31: A warm bias seems to occur in the south, maybe a map of SST differences would make biases straightforward for the reader.*

We will include a map of SST difference that indeed shows a positive bias of less than 1C in the southern part of the domain.

15) *Figure 1: Arrows on a) and b) are almost invisible.*

This will be corrected.

16) *Figure 2: Validation on annual field does not inform on the ability of the model to correctly simulate the upwelling occurring in the southern part of the domain at the winter-spring time of the year. I believe it would strengthen confidence on the simulation to add this component.*

Seasonal evaluation figures will be added for sea-surface temperature, sea surface cholorophyll mixed layer depth as well as vertical sections of temperature, salinity and nitrate (see also our response to comments 4 and 25 by referee #1).

*17) L6-7, page 11: Why not telling here why you chose 70 m depth as upwelling criteria rather than explaining the reason much later.*

We will move our justification for using the 70m depth here (page 11).

*18) L14-15, page 13: the description of the upwelling does not fit with the dynamic of the upwelling in the southern part of the domain (Ndoye et al., 2014, 2015, 2017; Capet et al., 2017)*

We will correct this statement to" "Similarly, the Ekman-driven upwelling in the southern subregion is restricted to the winter and spring (Pelegri and Benazzouz, 2015; Capet et al., 2017)"

*19) L4, page 18: rather 300 km than 200 ?*

This will be updated to 300km.

*20) Section 6: NPP and regenerated production are calculated by the coupled model. Why authors use satellite-based models here?*

Data-based NPP estimates are used because the model domain covers only partially the NASE and NATR provinces. We will added new in-situ based NPP estimates from Tilstone et al. (2009) and a new satellite-based NPP product (Westberry et al., 2008) (please also see our response to previous comment (6)

*21) L31-34, page 31: Authors indicate some limitations of the biogeochemical model (absence of colimitation, absence of nitrogen fixation; l30-34, p31) but omit the potential role of different communities of phytoplankton. Indeed, the model used only represents a single community of phytoplankton, the representation of diatom type organisms (comprising a siliceous skeleton and likely to contribute significantly to the export of organic matter) could influence the export in the model. In terms of export, it has also been shown that the alternation of phases of intensification and relaxation of the upwelling favorable winds is important for the dynamics of the upwelling systems (blooms and sedimentation). The use of a climatological wind in this study is likely to play a role in the results because they don't represent these alternations.*

We acknowledge two additional model limitations in the caveat section: 1) the fact that the model does not represent multiple phytoplankton groups and 2) the use of climatological winds lacking high-frequency variability that may lead to a misrepresentation of some aspects of the complex upwelling dynamics. Yet, the fact the model is able to represent the observed large-scale distribution of nitrate and its seasonality (in the original and new validation figures above) suggests that the impact of these limitations on the study conclusion is limited. It is also important to note that when it comes to the issue of offshore transport, the specific nature of the phytoplankton community is of secondary importance. What matters much more is the role of dissolved organic matter, whose production could be related to phytoplankton community structure, but likely only weakly so. Please see our response to previous comment 2.

*22) Affirmation lines 33-34 is true on an annual basis but could be false during the monsoon season when subtropical warm depleted open ocean waters invade the shelf in the southern part of the Canary Upwelling System.*

As our focus is on the annual-mean nitrogen transport we keep this statement.

*23) L16-20, page 32: I agree that the western section is much more extended than the northern and southern exits but the role played by the West Africa Boundary Current (Mauritania Current here) plays a quite important role in cross-shore exchanges, it should be taken into account.*

We answered a similar comment made by Reviewer #1 (please see response to comment 25 by the first referee).

*24) L16-19, page 33: The final conclusion states that this study emphasizes the need for improving the resolution of eastern boundary currents in global coarse resolution models. I think it would be fair to cite at least Large and Danabasoglu (2006) who stressed this point 15 years ago.*

We will cite Large and Danabasoglu (2006) as a previous study pointing towards a similar conclusion.

---

## Author Response (AR1)

The authors are grateful for the thoughtful comments given by the two anonymous referees on our paper. Below, we respond to each point raised by the reviewers and explain the changes we've made to the manuscript accordingly. The reviewers' comments are shown below in italics writing while our response is indicated in a red font.

**Anonymous Referee #1**

We are grateful for the first referee's efforts in reviewing the manuscript. We believe our paper has significantly improved as a result of his/her comments. Below, we highlight our responses to the general and specific comments and explain the revisions we've made to the paper accordingly.

General comments

*The paper by Hailegeorgis et al. examined upwelling and transport patterns in the Canary Current System using a Lagrangian modeling approach. The authors identified latitudinal variability in water and nitrogen export, determined transit time of water parcels from the coast to the oligotrophic oceanic region, examined the role of major capes as drivers of upwelling and offshore transport, and quantified the coastal upwelling contribution to the nitrogen stock in the North Atlantic Tropical Gyral and the North Atlantic Subtropical Gyral East. Overall, I think that the paper does an interesting contribution to the understanding of nitrogen export patterns in the Canary Current System. However, important changes need to be made in order to recommend publication.*

*My main concerns are two:*

*1) The paper is not easy reading:*

*There are a lot of results and figures with multiple panels and supplement figures, which made fuzzy the paper's main points. There is some redundancy in the reported results. Subregional patterns not always showed important differences, so not sure if you always need reporting all subregional results in the paper main body (you can move some Figure's panels to the Supplement).*

In order to solve the redundancy and lengthiness issues in the main paper, we have merged some subsections as well as moved some figures/panels to the supplementary. More specifically:

- We have moved the model validation (including Figs. 1, 2 and 3, which are now Figs. S1, S2 and S16) into the Supplementary section following the reviewer comment 6.
- We have merged subsections 3.1 and 3.2 so there is now no subsection in Section 3.
- We have merged Figure 5 and Figure 6 in one single figure (now Figure 2) and moved some of their sub-panels to a figure in the supplementary (now Figure S22, SI).
- We have merged and consolidated subsections 4.4 and 4.5 into one subsection 4.4 titled "Structure of offshore transport", which includes results on depth structure and horizontal

structure of the offshore transport. We have also brought a figure on the depth structure of offshore transport from the supplementary material to this subsection (now Figure 6).

- Figure 9 has been moved to the supplementary (now Figure S23, SI) as suggested by the reviewer (see below).
- Figure 10 panels c-h has been moved to the supplementary (now FIgure S24, SI).
- We have merged Figure 10 (panels a-b) and  Fig 11 into one figure (now Figure 7).
- We have moved most of Section 7 including Table 5 (now Table S4) to the supplementary section, while we kept in the main paper a brief summary of the comparison between our results and previous studies.

*2) The manuscript length could be reduced, trying a better integration of the paper results. Results and discussion were mixed, which did not help to get the present study contribution.*

We have significantly shortened the paper and consolidated multiple sections (please see our response to previous comment).

*I am not an English native speaker, but below (specific comments) I made a series of suggestions to help making sentences more concise and clear.*

*3) The authors need explaining much clearly how all the Lagrangian patterns were calculated in the Method section.*

We have moved some important details of the Lagrangian experiment from the supplementary text into Section 2.2.

We have also improved the description of the Lagrangian experiment by adding a schematic to the supplementary (now Figure S18, SI) that summarizes the different steps followed to quantify the Lagrangian offshore transport (also shown below).

| Step 1 | Step 2 | Step 3 |
|---|---|---|
| **Transport driven particle release (ARIANE Quantitative Experiment)** | **Particle tracking (ARIANE Qualitative Experiment)** | **Upwelling identification** |
| The particle release strip is set 50km westward of the 200m bathymetry isobath to be the western boundary of the coastal upwelling domain | Particle trajectories are initialized at the initial conditions in the quantitative experiment (Step 1) | All particles that cross 70m depth (the base of the euphotic zone) upwards while still in the coastal domain (before exiting the coast) are considered **upwelled particles** |
| Particles are released at each vertical cell along the particle release strip on each day of one year of the experiment according to the coastward transport, where each particle is associated with 0.01 Sv of transport or less | Trajectories are tracked for 720 days or until a particle leaves the experiment domain (whichever happens first) – we call this **exiting the domain**. This includes the whole experiment domain as opposed to the coastal upwelling area in the quantitative experiment. | For all upwelled particles, their trajectories (obtained in Step 2) are considered from their upwelling until they exit the domain |
| Particle trajectories are terminated at 720 days or when they exit the coastal upwelling area (whichever comes first) – we call this **exiting the coast**. This domain is limited to the area between the particle release strip and the West African coast. | Trajectory information (incl. position, velocity, nearby tracer concentration) is saved on a 1-day interval | In order to account for an upwelled water mass that is double-counted (see Steps 1 & 2), if an upwelled particle leaves the coastal domain but later re-enters and re-upwells, its trajectory since its second upwelling will be discarded (the particle's time of exiting the domain is shortened). This is because the subsequent trajectory will have been considered as an upwelling of a separate particle released at the water mass's re-entry into the coastal domain. |
| For each particle, its associated volume, position, time, velocity and nearby tracer concentration are saved at its entry into the coastal domain and on its exit from the coast | In the case of a double-counted water mass (see Step 1), it is represented by two particle trajectories starting from its re-entry into the coastal domain | |
| Note that there may be **double-counting** of a water mass that occurs if a water mass enters the coastal domain (initiating release of one particle), exits the coast and re-enters (initiating release of a second particle) | | |

We then added details on ARIANE's quantitative and qualitative experiments from what used to be in the supplementary text. As part of the quantitative experiment, we explained how we save the initial condition (including associated volume) of a particle release and mentioned many important details including the particle release period of one year, the release of particles over all the cells along the particle release strip, the maximum volume associated with particles, etc (please see lines 10-20, page 7). As part of the qualitative experiment, we explained details including the maximum length of trajectories and the frequency at which they are saved, the information saved about each particle along its trajectory and the portion of particles that leave the experiment domain early (please see lines 21-28, page 7).
We also indicated in this subsection that ARIANE neglects vertical mixing in its computation of particle trajectories (please see lines 2-3, page 7). Finally, we have added a new plot (now Figure 1b) of trajectories of a small sample of particles, in order to visually illustrate the Lagrangian experiment.

*4) Model validation:*

*The authors need to include vertical sections and vertical profiles from the modeled variables and compare them with observations (WOA sections would be OK). I would like to see whether the vertical patterns in the model outputs have any significant bias. Is the nutricline depth consistent with observed patterns across the model domain? Is the model reproducing well the seasonal*

*variability in vertical patterns? It is not enough the correlation analysis in Fig. 1 from the Supplement, since well-correlated variables can have important differences in terms of magnitude.*

We agree with the reviewer. We have produced validations of vertical cross sections of temperature, salinity and nitrate (WOA) for each subregion and for each season (from Figure S6 to Figure S14 in the supplementary).

The model generally does a good job in reproducing the vertical variability in observations for all three variables across the different seasons. We have presented our evaluation with more detail under Model Evaluation, which is now presented in the supplementary material.

As examples, the figures below show seasonal validations of vertical cross sections of (from left to right) temperature at 16N, salinity at 25N and nitrate at 30N. Each of these sample plots is taken from a distinct subregion in our experiment.

[Figure]

*5) Since a correlation analysis per se does not show a potential bias in the simulated variables, besides the annual mean patterns shown in Fig. 2. I would like to see a comparison for mean seasonal patterns of SST and surface chlorophyll.*

We have produced seasonal validation maps of SST (Pathfinder) and chlorophyll (SeaWIFS) shown in Figs. S3 and S4 (SI), respectively (shown below, in that order).

[Figure]

Furthermore, we have added a comparison of modeled mixed layer depth (MLD) to observations in all seasons in Figure S5, SI (please see our response to comment #20)

*6) To reduce paper length, I recommend including the model validation as an independent section in the Supplement.*

Following the reviewer's suggestion, we have moved the model validation to the supplementary.

Specific comments:

*7) Please, refer to the supplement figures as Figure S1, Figure S2, etc. It was confusing when a paper and supplement figures were mentioned at the same time. As example, instead of using (Fig. 7 and Fig. 8, SI), use (Fig. 7 and Fig. S8).*

We have made these changes.

*8) Abstract*
*Pag.1, L11: "Our model analysis suggests that the vast majority of the upwelled waters originate from offshore and below the euphotic zone (70m depth), and once upwelled remain in the top 100m". I understand what you mean, but the statement is not clear. Consider that you defined upwelling as a water parcel crossing the 70 m depth level.*

The sentence in question has now been removed from the abstract.

*9) Introduction*
*Pag.2, L13-14: The sentence "leading to substantial modifications of the biogeochemical cycles there" is ambiguous. You could delete it.*

Done.

*10) Pag.2, L20: delete "potentially"*

Done.

*11) Pag.2, L32: delete "potentially"*

Done.

*12) Pag.3, L10: "surface jet associated with the upwelling flowing equatorward" => "surface jet associated with the coastal upwelling front, which flows equatorward"*

Done.

*13) Pag.4, L6: "and therefore did not estimate" => "so did not estimate".*

Done.

*14) Pag.4, L8-10: These sentences need additional work. Explain better but concise.*

When it comes to the quantification of the contribution of the Canary upwelling to the open ocean nitrogen budget, the Lagrangian approach presents a couple of advantages relative to the Eulerian approach. Because of its focus on water particle trajectories, Lagrangian tracking of water masses is better suited for the analysis of connectivity between the coastal and the open ocean regions. Furthermore, the Lagrangian method can be used to derive conditional statistics where subsets of particles that fulfill certain criteria are analyzed. This is useful for instance to restrict the analysis of offshore transport to upwelling particles only.

This is now better explained in the revised manuscript (please see lines 1-19, page 4).

*15) Pag.4, L13: What do you mean with "quantify the reach"*

We have modified this to 'quantify the offshore reach' (please see lines 6, page 4).

*16) Pag.4, L13: "the spatial structure and the dominant timescales"*

Done.

*17) Pag.4, L23-24: and quantifying the offshore export*

Done.

*18) Pag.4, L24: "We also investigate"*

Done.

*19) Methods*
*Pag.5, L2: Was bottom remineralization included in the model? This is a relevant source of nitrogen, which can largely influence inorganic nitrogen patterns on the shelf. If it was not considered, you should mention it as another limitation in the study.*

Yes, the sinking particulate organic matter that reaches the seafloor is  remineralized back into ammonium at a slower rate (0.003 d−1 ) than in the water column (0.03 d−1 for small detritus and 0.01 d−1 for large detritus). We have added that explanation and a reference to Gruber et al. (2006) (please see lines 7-9, page 5).

*20) Pag.5, L14-15: Did you use a monthly climatology for wind stress? If this is the case, vertical mixing was probably underestimated. Did you compare the simulated mixed layer depth with observations?*

We used a monthly wind climatology (QuickSCAT SCOW). We have added a comparison of the modeled mixed layer depth (MLD) with observations for all seasons (Figure S5, SI). Using a monthly climatology forcing likely contributes to the underestimation of EKE shown (now in Fig S1, SI). We have acknowledged this in the Implications and Caveats (Section 8, lines 21-26, page 25). However, much of the spatial and seasonal variability in upper ocean mixing is still captured by the model as can be seen in the mixed layer depth validation in Figure S5 (shown below). This is particularly true in the summer and fall when the simulated MLD agrees the most with observations. This has been mentioned in the Model Evaluation section in the supplementary material (please see lines 20-22, page 1, SI).

[Figure]

*21) Pag.5, L17-18: It was mentioned that the model was spun up by 12 years, and the study was based in simulation years 10 to 12. So does it mean that you used the last 3 years of your model spin up for the analysis? If this is the case, you have to report that the model was spun up by 9 years.*

Yes, we did spin up for 9 years then used years 10-12 to run the experiment. This correction has been made (please see lines 20-21, page 5).

*22) Figures 1 and 2: I suggest including the 200 m isobath (shelf break) as a contour line on the maps.*

This change has been made in the figures in the revised manuscript (see Figs S1 and S2, SI).

*23) Pag.9, L2: It is Figure 2 not 1.*

Done.

*24) Pag.9, L14: Indicate that vertical mixing is not considered in the Lagrangian analysis*

This has been clarified (please see lines 2-3, page 7).

*25) Pag.9, L25: Since the dominant circulation pattern in the CanCS appears to be along-shore, I am wondering why only oceanic particles were considered in the Lagrangian experiment. I would expect that upwelled particles also come from the northern and southern boundaries. Could the poleward undercurrent be a source for upwelled waters?*

It is true that potential upwelling driven by waters entering the coastal zone at 14N and 35N were not included. We have acknowledged this among the study's caveats (please see lines 27-28, page 26).

Our choice to only allow open ocean particle releases was dictated by technical constraints in Ariane that prevent individual segments to be considered simultaneously as entry and exit sections. Yet, in order to quantify the potential error associated with our simplification we have run a separate experiment where particles were allowed to enter the coastal ocean from the southern and northern boundaries of the upwelling strip. Our analysis reveals that we are missing only a very small amount of water by our choice, i.e., only about 1% and 3% of the total volume tracked in this study come through the northern and southern boundaries, respectively.

Furthermore, given the high rate of recirculation near the coast, a vast proportion of the particles that may have entered the coast alongshore, once upwelled and exported offshore, are likely to

return to the coast from the open ocean, in which case they would be sampled in our particle release.

We conclude that discarding particles entering the coastal ocean from the southern and northern boundaries of the upwelling strip is likely to cause only a limited error in our quantification of the offshore transport.

We have further highlighted this caveat and discussed its potential implications in Section 8 of the revised manuscript based on the arguments developed above (please see lines 28-34, page 26).

*26) Results*
*Pag.13 L7: "onshore-offshore contrast" => "cross-shore differences"*

We have made this change.

*27) Pag.13 L9: "followed by a second one at around 50 km from the shore" => "and a secondary maximum around 50 km"*

We have made this change.

*28) Pag.13, L20. Shallow regions should also show upwelling because a distance of 50 km from the shelf break was considered for the analysis. Right? Clarify.*

The reason there is no upwelling at the very coast in these regions is that the bathymetry is shallower than 70m, therefore making it impossible to meet our criteria for upwelling where a particle crosses the 70m depth. The minimum bathymetry allowed by the model is 50 m and that has been added to the model's description in section 2.1 (Page 5, Lines 12-13) as Reviewer 2 has also requested (see below). However, as mentioned by the referee we sample upwelling further offshore in these locations. Therefore, for more clarity we have changed the statement to: "our experiment identifies limited coastal upwelling south of Cape Blanc and between Capes Barbas and Bojador because their bathymetry is shallower than the 70 m upwelling depth criterion used here" (please see lines 21-2217, page 9).

*29) Pag.13, L25: what is the range considered for the upwelling zonal integration in Figure 6a (same for Figure 5a).*

We integrate zonally across the coastal stripe (Between the coast and  50 km westward of the 200m bathymetry). We have added this clarification to the caption for the panels (now Figure 2).

*30) Pag.13, L26-28: Wondering if the shallow nutricline is linked to the high-nutrient SACW or not.*

No. The shallow nutricline is a consequence of the shallow thermocline which is caused by the stronger stratification in the tropics in comparison with the high latitudes.

*31) Pag13, L28-29: "The central subregion has a moderate nitrogen flux associated with upwelling while the northern subregion has the weakest upwelling flux of nitrogen" => "The weakest upwelling-driven nitrogen flux was in the northern subregion"*

We have made this change.

*32) Pag.13, L30: "disproportionately lower" => "much lower"*

The sentence in question has now been removed in the revised manuscript.

*33) Pag.15: It is important to note that a water parcel released in region A can be upwelled in subregion B or C. Beside, a water parcel upwelled in region A can be transported away to region B or C. Consequently, the parcel transport not necessarily depends on the oceanographic conditions in region A. If I'm right, I suggest revisiting all sentences describing subregional offshore transport results. As example, I would modify "the offshore transport is fastest in the central subregion and slowest in the northern subregion" by "the particle upwelled in the central and northern subregions displayed the fastest and slowest offshore transport, respectively". Besides, I would change "At larger distances from the coast (beyond 400 km), the offshore transport becomes faster in the southern subregion" by "The fastest water parcels beyond 400 km from the coast are those upwelled in the southern subregion"*

We have followed this suggestion and corrected the way we express transport from each subregion accordingly. This has been applied throughout the paper.

*34) Pag.15, L3: define "transit time".*

We have added here the transit time definition from the caption of Figure 3 in the revised manuscript.

*35) Pag.15, L15-16: Why is this maximum at 150 km? Does it mean that a greater fraction of water parcels remains around this distance? Is this linked to patterns in alongshore circulation?*

In the revised manuscript, we have added an explanation that the distance of maximum transport corresponds to the distance reached by most sampled upwelled water (nitrogen). It is around 150km (that roughly corresponds to the width of the upwelling stripe) because at closer distances to the coast (<100-150km), upwelled volume is only partially sampled. Farther distances (>100-150km) are never reached by a proportion of particles because of recirculation retaining them close to the coast and alongshore transport exporting some particles out of the model domain (please see lines 5-10, page 11).

*36) Pag.15: L22: "This is because of its low offshore transport efficiency and the relatively low volume of upwelling" => ''Reduced coastal upwelling and low offshore transport efficiency explain this pattern."*

We have made this change.

*37) Pag.15, L22: Define transport efficiency.*

Transport efficiency (the ratio of offshore transport volume at a given distance to upwelling volume) has been defined in the revised manuscript (Page 11, Line 16).

*38) Table 2: It may be informative reporting the standard deviation for transit time.*

We have replaced Table 2 with a graph of transit times featuring error bars (Figure 3 in the revised manuscript).

*39) Pag.16, L3-5: "At close to 150 km from the coast (corresponding to the edge of the coastal upwelling stripe), the offshore transport of nitrogen reaches its maximum, reaching values as large as 500 Gmol yr−1. Thereafter, the offshore transport decreases exponentially" => "Around 150 km from the coast, the nitrogen transport reaches values as large as 500 Gmol yr-1 at 200 km, decreasing exponentially further offshore"*

We have made this change.

*40) Pag.16, L7: "In spite of having the lowest offshore transport of water, the southern subregion exports the highest amount of nitrogen offshore at 200 km" => "Although offshore water transport was minima in the southern CanCS, this subregion has the greatest offshore export of nitrogen."*

We have made this change.

*41) Pag.16, L9: "At larger distances from the coast, the situation reverses." => "This pattern reverses further offshore"*

We have made this change.

*42) Pag.18, L2-4: I disagree with this statement: "so that beyond the nearshore 50 km region, inorganic nitrogen in the form of nitrate dominates the nitrogen pool at all distances from the coast". NO3 dominates almost everywhere, and its contribution to total nitrogen is actually much larger in the coastal region than in the oceanic region.*

We agree with the referee that $NO_3^-$ dominates nearly everywhere. Yet, in the nearshore 50km region, the sum of all organic nitrogen (in both plankton and detritus forms) can exceed that of

nitrate as shown by the fraction of organic N in the offshore transport (red dashed curve) in Figure 5.

*43) Pag.18, L21-34: It would be nice having a brief introductory explanation for the motivation of the analysis in this subsection. As example, evaluate the impact of subduction on transit time, impact of subduction on nutrient fluxes or nutrient cycling.*

We have added introductory statements in the beginning of this subsection on the potential impact of vertical structure of offshore transport (subduction) on transit times and offshore transport efficiency to serve as motivation for the depth structure analysis (Page 14, Lines 5-7). Later in the subsection, these potential links are revisited based on the analysis that has just been presented (Page 16, Lines 3-5). In general, subduction increases transit times (slows down offshore transport) since velocities deeper in the water column are smaller. However, subduction can also increase the efficiency of offshore transport by minimizing the depletion of nitrogen in surface waters.

In the original draft, we had also related subduction of water particles with filament activity (page 18, lines 30-31) and with nitrogen offshore transport efficiency (Page 31, Lines 18-23; Page 32, Lines 2-4).

*44) Pag.18, L21: "Upwelling particles that..." => "Upwelled particles that..."*

We have made this change.

*45) Pag.18, L22: How do patterns in Fig. 8a,b from the supplement were calculated?*

To make Figure 8 a (or b) in SI, we first considered all downward (or upward) movements of all particles across the horizontal plane at 50m depth. After finely binning the horizontal surface across the model, we added the associated volumes of all particles that crossed this depth downward (or upward) for each binned area throughout the 2 year particle trajectory experiment. We have added more detail to the caption of the figure (now Fig. S27) in the Supp. Information to show this more clearly.

Although Fig. 8c in SI gives the dominant vertical transport (upward or downward) in the modeled region, panels (a) and (b) also complement this information by giving the full picture of where the upward and downward transport occurs during the whole 2 year experiment. Panels (a) and (b), for example, show that although the coast is dominated by an upwelling flux, it also features significant downwelling.

*46) Pag.18, L24: I cannot see the secondary upwelling in the open ocean. Describe better.*

It is true that Figure 10d in the submitted manuscript doesn't show significant upwelling flux for particles upwelled in the southern subregion except for a subsurface upward transport maxima in the first 300km from coast. So we now only cite Figure S27(c) in the revised supplementary material, which does show strong net upward transport (upwelling) in the open ocean in the southern CanCS, which has significant overlap with the trajectory of the upwelling from this subregion.

*47) Pag.18, L30: It is not evident for me why persistent filaments contribute to enhanced subduction. Explain.*

Here, we were referring to the role of filaments in squirting cold upwelled water offshore and the subsequent subduction of this cold water due to its high density compared to the open ocean's warmer surface water. We have now added a sentence stating this fact including a reference to Lovecchio et al. (2018), who found filaments to cause subduction of organic nitrogen, for example, to depths larger than 100m (Page 14, Lines 13-15, in the revised manuscript).

*48) Figure 9: I recommend include this figure in the Supplement.*

We have made this change.

*49) Pag.20. Indicate what the positive/negative values in Fig. 10 represent*

We have made this change.

*50) Pag.21. Indicate what the positive/negative values in Fig. 11 represent*

We have made this change.

*51) Pag.22, L26: "offshore transport of upwelled particles"*

We have made this change.

*52) Pag.22, L30: I am not sure whether remotely upwelled water is a good term. I would prefer describing the results in terms of local and non-local upwelling.*

We have changed all references to "remote upwelling" to "non-local upwelling".

*53) Pag. 22, L32: "water upwelled"*

We have made this change.

*54) Pag. 23: L1: "Corresponding enhancement in local nitrogen upwelling and export is seen only in" => "Increased offshore transport of nitrogen due to increased local upwelling is seen only in"*

We have made this change.

*55) Caption of Figure 12: "Transport by water upwelling locally" => "Transport associated with locally upwelled water"*
*"Transport by particles that leave the coastal upwelling region at each cape or non-cape area but upwell remotely" => "Transport associated with remotely upwelled water"*

We have made these changes. We have slightly modified the second edit suggested by the referee to "Transport associated with non-locally upwelled water at each cape or non-cape area".

*56) Pag.24, L1-2: "The offshore transport of nitrogen by remote upwelling exported by all capes constitutes more than 30% of the total offshore transport of upwelling" => "Remotely upwelled waters that are transported offshore around major capes represent more than 30% of the total transport"*

We have made this change.

*57) Pag.24, L4-5: "In fact, all capes source the majority of water and nitrogen they export from remote upwelling (Table 3). All capes also source more of their export from remote upwelling compared to the rest of the coast." => "Indeed, most of the water and nitrogen exported offshore around major capes is non-locally upwelled (Table 3)"*

The second sentence is meant to show the high export of non-local upwelling that occurs at capes compared to the non-cape coast. But the phrasing was unclear so we have modified it slightly to: "Each cape also sources more of its export from non-local upwelling than any of the non-cape coastal areas." (Page 17, Line 21).

*58) Pag.27, L10: For consistency use CanCS*

We have made this change.

*59) Pag.27, L21-29: I did not understand. Please, explain better how did you estimate the CanCS contribution to the NATR and NASE provinces.*

We have added a brief clarification of our calculation to the footnote of Table 3 in the revised manuscript. Briefly, it is as follows.

We explain that whenever a particle enters a province within the top 100m, the nitrogen it carries into the province is added to the particle's source subregion's contribution to the province. On the contrary, when a particle from a given subregion leaves a province within the top 100m, the nitrogen it carries with it when it leaves the province is subtracted from the contribution of the subregion of the particle to the province.

We use precise Longhurst province boundaries for our analysis. Analysis of crossing into and out of a province is determined based on a daily resolution of particle positions since that is the resolution of our Lagrangian trajectories.

[[[ [[ ///// \\\\\

**Anonymous Referee #2**

We would like to thank the second reviewer for the comments that have, we hope, significantly improved our manuscript. Below, we highlight our responses, point by point, to the reviewer's general and specific comments and indicate the revisions we have made to the paper accordingly.

General comments

*This study aims to determine the contribution of nitrogen upwelled within the coastal region of the Canary Upwelling System to the nitrogen budget of the open ocean through a Lagrangian study relying on model outputs generated by a coupled physical-biogeochemical experiment. Authors also aim at describing the timescales, the reach and the structure of this offshore transport to quantify the role played by upwelling on nitrogen enrichment of the NATR and NASE provinces as defined by Longhurst. I'm not sure the study makes a significant contribution to the issue of nitrogen irrigation of the NATR and NASE provinces. There are several reasons for this.*

1) *First, the authors justify the originality of their study by the use of a Lagrangian approach as opposed to Eulerian approach which has been used in Auger et al. (2016) or Lovecchio et al. (2017, 2018) which have been mentioned in the present study. The authors in particular advance the capacity of their Lagrangian approach to define more faithfully the contribution of the coastal upwelling region in terms of nitrogen supply to the subtropical gyre (l5-10, page 4). This statement seems relevant with regard to the volume transported from the coastal region to the open sea, but much less obvious with regard to the transport of nitrogen. Indeed, the amount of nitrogen carried by each particle to a given location is quantified as the product of the particle associated volume and the concentration of the tracer associated with the particle when it reaches that location (l4-6*

*page 11). To my understanding of the methodology, the nitrogen concentration at a particular location does not necessarily come from the coastal upwelling but can be supplied locally, can change its chemical form or have a different origin.*

We do agree with the referee that in contrast to water volume, tracing the transport of nitrogen is somewhat more difficult given the chemical transformations between inorganic and organic nitrogen. In addition, subgrid mixing is not represented in our Lagrangian particle tracking but can affect nitrogen concentrations in ways that are not accounted for in our transport estimates.

Yet, as has been shown also by Frischknecht et al. (2018), the Lagrangian method permits a lot of new insight into the offshore transport of nitrogen, since total nitrogen, i.e., the sum of inorganic and organic nitrogen is conserved except for the part that is sinking, and that part that is being supplied through mixing. Indeed, along the way, nitrogen can be incorporated into organic matter and then being recycled again, but if it is tracked by our algorithm, this nitrogen is still coming from the coastal upwelling.

The component we lose through sinking does not affect our conclusions, since this component is lost to the ocean interior, from where it will not find its way back into the waters that are transported offshore. More importantly is our lack of consideration of the vertical mixing. We have good evidence that this component is relatively small. First, the total amount of nitrogen is decreasing with offshore distance, and not increasing. In fact, the decrease is driven entirely by the sinking component, and the spatial distribution of this loss fits well the spatial distribution of the export of organic nitrogen (Figure 4 in the revised manuscript). In particular, we see a decline in total nitrogen as a function of distance to the coast that is sharper than for water volume. And this decline is larger for water particles originating from the southern subregion that are transported at the shallowest depths (Figure S23 in the revised SI). If the supply of nitrogen from surrounding waters to upwelling waters due to mixing were large enough to cancel the loss due to organic matter sinking, there would be no such a sharp decline in the offshore transport of nitrogen as a function of distance to coast. This suggests that although the potential changes in nitrogen due to subgrid mixing can locally be important, they are unlikely to affect the large-scale transport estimates in a significant way.

Yet, we acknowledge that the lack of a representation of mixing in Ariane is an important caveat that not only can affect particles' depth, but also potentially their nitrogen content and hence locally our offshore transport estimate. This is now explicitly stated in the discussion of the method caveats (Section 8, Page 25 (Line 34) to Page 26 (Line 16)).

2) *In addition, authors indicate some limitations of the biogeochemical model (absence of colimitation, absence of nitrogen fixation; l30-34, p31) but omit the potential role of different communities of phytoplankton. Indeed, the model used only represents a single phytoplankton community, the representation of diatom organisms (comprising a siliceous skeleton and likely to contribute significantly to the export of organic matter) could influence the export in the model. In*

*terms of export, it has also been shown that the alternation of phase of intensification and relaxation of the upwelling favorable winds is important for the dynamics of the upwelling systems (significant efflorescence generation and sedimentation). The use of climatological wind in this study is likely to play a role in the results because it does not represent these alternations. These aspects should be mentioned in the limitations of the study.*

We agree that similar to other state-of-the-art models, our model (especially the biogeochemical module) has other limitations beyond what we have already acknowledged in the original version of the manuscript. Yet, the fact that the simulated (1) distributions of nitrate and its seasonality (Figs S12, S13, S14) and (2) the POC vertical profile (Fig S15) agree relatively well with the observations suggests that the impact of these limitations on the study conclusions is likely limited. See also our response to comment 4 by Reviewer 1.

Nevertheless, following the referee's suggestions we have added two additional potential model limitations in the caveat section (Section 8): 1) the fact that the model does not represent multiple phytoplankton groups and 2) the use of climatological winds lacking high-frequency variability that may lead to a misrepresentation of some aspects of the complex upwelling dynamics (please see lines 21-26 and lines 29-31, page 25).

3) *Then, the conclusions of the study highlight the importance of the Capes in the generation of filaments which represent privileged export sites but the influence of topographic accident on the generation of filaments has already been studied theoretically (eg Meunier et al., 2010), through hydrodynamic simulations and observations for certain filaments of the Canary upwelling system. The quantification of the overall contribution of filaments and the extension of the source waters supplying the main filaments of the system nevertheless provides interesting information, even if the three-dimensional dimension of upwelling is becoming more and more essential in the literature targeting these upwelling regions.*

We agree with the referee in that previous studies like Meunier et al. (2010) and Troupin et al. (2012) have demonstrated the importance of coastal topography and capes in particular in the formation of coastal filaments. Therefore, we have added references to these previous works in the revised manuscript.

In particular:
- we have changed the statement in the original manuscript:"Coastal filaments along the West African coast can occur everywhere, but it is well established that the majority of the filaments are persistently associated with the major capes along the coast" (Page 22, Line 14) to "Coastal filaments can occur everywhere anywhere on the coast in the CanCS, but previous studies have shown that capes can facilitate their formation (Meunier et al., 2010; Troupin et al., 2012)" (Page 16, Lines 23-24 in the revised manuscript).
- to the list of citations on page 24, line 10 in the original manuscript, we have added Meunier et al. (2010) and Troupin et al. (2012) to highlight that the alongshore advection can

interact with capes to result in the formation of a coastal filament that then exports upwelled water to the open ocean (Page 17, Lines 27-29 in the revised manuscript).

4)  *The role of mesoscale activity on residence times and the kinetics of transport from the coastal zone to the open sea is also part of the presented results. Mesoscale activity in the transition zone has been widely studied in all eastern boundary upwelling systems and fairly exhaustively in the northern part of the Canary system, in particular from the ROMS model (Mason et al., 2011 & 2012 ; Troupin et al., 2012).*

We agree that previous studies mentioned by the referee have studied the mesoscale variability in the northern Canary system. We have added references to these papers in the revised manuscript.
In particular, we now cite:

- Mason et al. (2011) in section 5.2 among the papers we cite to refer to mesoscale variability and overall transport complexity in the Canary (line 1-3, page 20).

- Mason et al. (2012) in our literature review in the introduction section and section 5.2 on recirculation (line 22-23, page 4; line 1-3, page 20).

- Troupin et al. (2012) in our description of the Cape Ghir filament (line 25-29, page 16).

- Barton and Aristegui (2004) on mesoscale activity in the Canary (see subsection 5.2, line 1-3, page 20).

5)  *In the southern part of the area studied, the underestimation of EKE (Figure 1), an activity also highlighted by the occurrence of eddies in this region (Schutte et al., 2016), is not mentioned and is likely to impact the results in this region. The literature on the region is also to be completed, in particular to take into account recent studies by German, Senegalese and French teams. This update particularly concerns the southern part of the system which would allow the authors to describe their results more precisely. Hydrological conditions off Mauritania are described in Klenz et al. (2018), the vortex activity is studied in Schütte et al. (2016), the understanding of the dynamics of the Mauritanian current was revisited by Kounta et al. (2018), and the functioning of the Senegalese upwelling by Ndoye et al. (2014, 2015, 2017) or Capet et al. (2017). These studies point in particular to the importance of the Mauritanian current (to which I prefer the name West Africa Boundary Current; Kounta et al., 2018) on the dynamics of upwelling.*

We agree with the referee that the model underestimates the EKE  in the coastal area of the southern subregion, and particularly so south of Cape Verde. Following the referee's comment, we now explicitly mention the underestimation of the EKE in that region and its potential implications among the study's caveats (Section 8, line 21-26, page 25).

Following the referee's suggestion, we also have expanded our review of the literature in the region. In particular, we have cited the following works in the revised manuscript:

- Schutte et al. (2016) and Kounta et al. (2018): are cited in subsection 5.2 (as well as in the Introduction section) to highlight the potential importance of the Mauritanian current and eddies in the region in fueling offshore export of water (please see lines 9-11, page 3; line 1-3, page 20; lines 5-6, page 21).
- We also cite Glessmer et al. (2009) and Peña-Izquierdo et al. (2015) to emphasize the importance of the Mauritanian current in the Introduction section (please see lines 9-11, page 3).
- Klenz et al. (2018): is now cited in section 5.1 to highlight the importance of both the poleward (Mauritanian) current as well as the equatorward Northern Atlantic (Canary) current, particularly during winter, as a source of upwelling in the southern subregion. This helps explain the patterns in Figure 13 (now Figure 9 in the revised manuscript) for the source waters of exports at capes in the southern subregion (please see line 31-32, page 17).
- Ndoye et al. (2017) and Capet et al. (2017): have been cited in our description of the Cape Verde filament and the importance of the local mesoscale activity in section 5.2 (please see lines 25-29, page 16; lines 1-3, page 20; lines 5-6, page 21).

6) *Finally, questions remain as to how to assess the contribution of nitrogen from coastal waters to new production. Indeed, I did not understand the use of VGPM models to quantify primary production knowing the large differences that exist in satellite-based models of primary production in the region (Gomes-Letona et al., 2017).*

We used satellite-based (VGPM) NPP estimates because of their synoptic-scale coverage, which individual in-situ estimates lack. However, we are aware of the important uncertainties associated with this (and other satellite-based) product(s). Therefore, we have now added productivity estimates based on in-situ measurements that are available for some Longhurst provinces (Tilstone et al., 2009) as well as estimates from the CbPM model (Westberry et al., 2008).

Indeed, in-situ estimate of primary production for the NATR based on Carbon-14 uptake from Tilstone et al. (2009) has been added. This estimate (1377mmol N m-2 yr-1) is less than that derived from satellite data (1753 mmol N m-2 yr-1). This suggests that the contribution of the Canary upwelling nitrogen supply can be locally even more important in relative terms, than what our initial estimates have implied. Using a carbon-based productivity model (CbPM), NPP estimates for the NATR and NASE both amount to around 2000 mmol N m-2 yr-1. This also suggests that the CanCS's contribution to the NASE is locally more important than what the province's VGPM value (2139 mmol N m-2 yr-1) suggested.

We have updated Table 4 (now Table 3) and the discussion of the potential contribution of the nitrogen transport to the total NP estimates in the NATR and NASE provinces accordingly (please

see lines 10-13, page 23; lines 22-24, page 23). In the table, we now don't show the NPP values but only the NP value range derived as a product of these NPP values and the ef-ratios.

Finally, Gomez-Letona (2017) compares three alternative estimates of PP and finds divergences between them. However, their estimates and comparisons are focused on the coastal area while we were primarily interested in finding estimates for the larger adjacent Longhurst provinces so that we could compare offshore transport of nutrients in our model with the provinces' total budgets.

7) *The manuscript is however well written, well illustrated with clean and condensed figures, the methodology is well described, and the main messages are clearly presented.*

We thank the referee for his/her positive and encouraging comment.

8) *As a summary, the manuscript is of good quality but I hardly consider the results as really moving our understanding forward. The methodology that uses Ariane as a Lagrangian tool is supposed to make a difference in the description, quantification of nitrogen irrigation of NATR and NASE provinces but I'm not convinced that it solves the issues faced by an Eulerian approach.*

Again we thank the referee for praising the quality of the manuscript. We believe that while it comes with important limitations, our study does improve the quantification of the coastal upwelling supply of nitrogen to the open ocean, relative to the Eulerian approach. We detail our reasoning in our response to the previous comment #1 by the same reviewer.

Specific comments

9) *L21-23, page 2: Reformulate the sentence "Especially low-latitude ..." which is hardly understandable.*

We have made this change (please see lines 15-17, page 2).

10) *L12, page 3: The current of Mauritania must be considered in the light of the work of Kounta et al. (2018).*

Done. See response to comment 25 above.

11) *L30, page 4: Did you use ROMS or CROCO oceanic modeling system?*

We used ROMS-AGRIF version 3.1.1 (which shares the same code with the current version of CROCO).

12) *In this section 2.1.1, indicate the shallowest depth used at the coast (hmin parameter).*

We have now indicated the 50m lowest bathymetry in section 2.1 accordingly (please see lines 12-13, page 5).

13) L26-28, page 5: EKE in the southern part of the domain is underestimated, please tell it and justify it.

We indicated in the revised manuscript that the model underestimates EKE in the southernmost part of the CanCS region as well (Section 8, lines 21-22, page 25).

14) L30-31: A warm bias seems to occur in the south, maybe a map of SST differences would make biases straightforward for the reader.

We have included a map of SST difference that indeed shows a positive bias of less than 1C in the southern part of the domain (Figure S2, Supp. Info).

15) Figure 1: Arrows on a) and b) are almost invisible.

This has been corrected.

16) Figure 2: Validation on annual field does not inform on the ability of the model to correctly simulate the upwelling occurring in the southern part of the domain at the winter-spring time of the year. I believe it would strengthen confidence on the simulation to add this component.

Seasonal evaluation figures have been added for sea-surface temperature, sea surface cholorophyll, mixed layer depth as well as vertical sections of temperature, salinity and nitrate in Figures S3 to S14, SI (see also our response to comments 4 and 25 by referee #1).

17) L6-7, page 11: Why not telling here why you chose 70 m depth as upwelling criteria rather than explaining the reason much later.

We have moved our justification for using the 70m depth here (page 7, line 29-30).

18) L14-15, page 13: the description of the upwelling does not fit with the dynamic of the upwelling in the southern part of the domain (Ndoye et al., 2014, 2015, 2017; Capet et al., 2017)

We have corrected this statement to" "Similarly, the Ekman-driven upwelling in the southern subregion is restricted to the winter and spring (Pelegri and Benazzouz, 2015; *Capet et al., 2017*)" (please see lines 15-16, page 9).

19) L4, page 18: rather 300 km than 200 ?

This has been updated to 300km (please see line 21-22, page 12).

20) *Section 6: NPP and regenerated production are calculated by the coupled model. Why authors use satellite-based models here?*

Data-based NPP estimates are used because the model domain covers only partially the NASE and NATR provinces. We have added new in-situ based NPP estimates from Tilstone et al. (2009) and a new satellite-based NPP product (Westberry et al., 2008) (please also see our response to previous comment (6)).

21) *L31-34, page 31: Authors indicate some limitations of the biogeochemical model (absence of colimitation, absence of nitrogen fixation; l30-34, p31) but omit the potential role of different communities of phytoplankton. Indeed, the model used only represents a single community of phytoplankton, the representation of diatom type organisms (comprising a siliceous skeleton and likely to contribute significantly to the export of organic matter) could influence the export in the model. In terms of export, it has also been shown that the alternation of phases of intensification and relaxation of the upwelling favorable winds is important for the dynamics of the upwelling systems (blooms and sedimentation). The use of a climatological wind in this study is likely to play a role in the results because they don't represent these alternations.*

We acknowledge two additional model limitations in the caveat section: 1) the fact that the model does not represent multiple phytoplankton groups and 2) the use of climatological winds lacking high-frequency variability that may lead to a misrepresentation of some aspects of the complex upwelling dynamics (please see lines 21-26 and lines 29-31, page 25). Yet, the fact the model is able to represent the observed large-scale distribution of nitrate and its seasonality as well as the vertical mean POC profile (in the original and validation figures and Figs S12-S15, SI) suggests that the impact of these limitations on the study conclusion is limited. It is also important to note that when it comes to the issue of offshore transport, the specific nature of the phytoplankton community is of secondary importance. What matters much more is the role of dissolved organic matter, whose production could be related to phytoplankton community structure, but likely only weakly so. Please see our response to previous comment 2.

22) *Affirmation lines 33-34 is true on an annual basis but could be false during the monsoon season when subtropical warm depleted open ocean waters invade the shelf in the southern part of the Canary Upwelling System.*

As our focus is on the annual-mean nitrogen transport we keep this statement.

23) *L16-20, page 32: I agree that the western section is much more extended than the northern and southern exits but the role played by the West Africa Boundary Current (Mauritania Current here) plays a quite important role in cross-shore exchanges, it should be taken into account.*

We answered a similar comment made by Reviewer #1 (please see response to comment 25 by the first referee).

24) *L16-19, page 33: The final conclusion states that this study emphasizes the need for improving the resolution of eastern boundary currents in global coarse resolution models. I think it would be fair to cite at least Large and Danabasoglu (2006) who stressed this point 15 years ago.*

We now cite Large and Danabasoglu (2006) as a previous study pointing towards a similar conclusion (please see lines 24-25, page 27).

[revised manuscript text omitted]

---

## Author Response (AR2)

**The authors are grateful for the valuable comments given by the editor and the two anonymous referees on our revised manuscript. Below, we respond to each point raised by the editor as well as by the reviewers and explain the changes we've made to the manuscript accordingly. The editor and reviewers' comments are shown below in italics writing while our response is indicated in a blue font.**

Associate Editor Decision: Publish subject to minor revisions (review by editor) (21 Sep 2020) by Minhan Dai

*Comments to the Author:*
*Dear Authors,*

*I have now had your revised MS further evaluated by two reviewers. Both reviewers appreciated your efforts during the revisions. However, both reviewers also pointed out that your paper has room to improve in its presentation. In addition, and as per the comment from reviewer #2 regarding Lagrangian vs Eulerian approaches, please explicitly state in your MS the limitations of the Lagrangian approach.*

*Sincerely,*

*Minhan Dai*
*Editor*

We deeply thank the editor for handling our manuscript and for his comment.

Following the reviewers suggestions we have improved the presentation of the paper by shortening and better focusing the introduction as suggested by reviewer #1. We have also added some additional clarifications to the description of the Lagrangian experiment as requested by reviewer #1.

As for the Lagrangian approach limitations associated with the lack of an explicit representation of subgrid mixing, we would like to highlight the fact that we fully acknowledge this caveat in the discussion section (please see line 34, page 25 to line 21, page 26).

However, contrary to what Reviewer #2 comment may seem to suggest, we believe the bias associated with this to be small. In fact, given the high spatial and temporal resolution of our velocity field, there is effective vertical mixing occurring as a result of vertical advection and resolved mesoscale and submesoscale dynamics (e.g., secondary ageostrophic circulation around fronts) that is captured by the Lagrangian experiment. We also cite Frischknecht et al. (2018), who have found highly consistent budgets between Eulerian and Lagrangian frameworks they applied in their analysis of the California Current System. We now further stress this in the discussion (please see lines 11-15, page 26). We also have included the following additional statement in the introduction to further highlight these potential limitations of the Lagrangian approach (lines 30-33, page 3):

"However, a Lagrangian approach also comes with a number of disadvantages. Perhaps the most important one is the difficulty to fully take into account subgrid mixing that is associated with the model treatment of unresolved physics."
* * *
Report #1

Submitted on 02 Sep 2020
Anonymous Referee #1
* * *
*The manuscript by Hailegeorgis et al. substantially improved from it's first version. The authors did a good validation of the ocean-biogeochemical model and wrote a nice discussion (section 8), which helps to contextualize better the work contribution.*

We are grateful to the first referee's efforts in reviewing the manuscript. Below, we highlight our point-by-point responses to the reviewer comments and explain the revisions we've made to the paper accordingly.

*However, the paper still need some extra work. Specifically, the description of the model experiment and results still need clarification. This is highly relevant to capture the paper's take home messages. Besides, an additional effort should be done to improve the Introduction section, reducing wordiness and improving paragraph's flow. I would recommend five well written paragraphs instead of seven. Specific comments are indicated below:*

Done. We have significantly shortened and better focused the Introduction, reducing wordiness and dropping unnecessary or redundant information. In particular, we have reduced the Introduction to five paragraphs instead of seven initially (please see page 2, lines 7-11; page 2, line 33 to page 3, line 8; page 3, lines 31-33 ).

*Page 5: The authors provided a scheme to explain better the experiments, but this scheme seems to me a little bit confusing. I am wondering why they defined those 3 modeling steps. I also wonder why the second step was called "qualitative", although this step quantifies position, velocity, and tracer concentration. It may be helpful to state clearly the goal of each step in page 5. Besides, it could be helpful to explicitly link the steps with the results presented in sections 3-5. I suggest moving an improved experiment diagram to the paper main body.*

The quantitative and qualitative experiments referred to in the initial revision of the manuscript are names used in ARIANE to describe two types of Lagrangian experiments, that either only save the initial and final positions and volumes associated with particles crossing some predefined entry sections (Quantitative) or the

entire trajectories associated with each particle (Qualitative). In our case, we use the former to identify the location, time and volumes of particles at their release/seeding and the latter to track them along their trajectories. We agree with the referee that this naming may be a source of confusion for people not familiar with the ARIANE jargon. Therefore, we dropped these names and focused on the goal of each step, as also suggested by the referee. We have modified the labels of the three stages of the experiment (page 5, lines 20-21) and edited the names of the stages in the subsequent description accordingly.

As we have clarified in the text the three steps involved in the Lagrangian modeling experiments, we have dropped the schematic in the supplementary information as it mostly brings redundant information. We have, instead, added a brief clarification of the steps in Figure 1c.

*Page 6: Suggestion: "Characteristic of upwelling" => "Upwelling patterns"*

Done.

*Page 9: Suggestion: "But the offshore distribution of the upwelling" => "But the upwelling zonal distribution"*

Done.

*Fig S22: Do these patterns represent the distribution of (1) upwelled particles at a given time, or (2) the locations where particles were upwelled above 70 m depth? Please clarify.*

The patterns represent the location where upwelling occurs. We have changed the caption to "Zonal distribution of upwelling volume (a-c) and nitrogen content (d-f)…"

*Page 10: Suggestion: "the median time for particles to traverse a given offshore distance interval" => "the median particle's travel time for a given offshore interval"*

Done.

*Page 10: I am wondering what was the impact of islands in the offshore transport of water and nitrogen. This is an aspect that was not discussed in the paper.*

The role of islands is not explicitly studied in this work. However, in our discussion of the role of recirculation we refer to the role of eddies entrained by the Canary Archipelago (Canary Eddy Corridor) as a potential contributor to the prominence of the indirect transport of upwelled waters in the central and northern subregions. Please see (page 20, line 16 to page 21, line 4):
"The importance of the indirect transport of upwelled waters in the central and northern subregions can be linked to the prominent role played by mesoscale eddies there. Indeed, eddies with length scale of 100 km to 300 km are known to be important at these latitudes (Mittelstaedt, 1991). These include a recurrent cyclonic

eddy south of Cape Juby and the cyclonic and anticyclonic eddies entrained by the Canary Archipelago, forming the so called Canary Eddy Corridor (CEC), which is located at 22º –29º (Arístegui et al., 1994; Piedeleu et al., 2009; Sangrà et al. 2009). This region of long-lived westward-propagating eddies is known to contribute strongly to the offshore transport of organic matter and carbon (Sangrà et al., 2009)."

*Page 11, line 6: I'm lost, why is that coastal upwelling area the most sampled?*

Done.

*Page 14: Suggestion: "4.4 Structure of offshore transport" => "4.4 Vertical and horizontal structure"*

Done.

Report #2

Submitted on 07 Sep 2020
Anonymous Referee #2

We are grateful to referee #2 for his/her efforts in reviewing the revised manuscript. Below, we highlight our point-by-point responses to the reviewer comments and explain the revisions we've made to the paper accordingly.

*I am still not completely convinced of the level of added value brought by the Lagrangian approach compared to the Eulerian approach. In their response, authors acknowledge the "lack of consideration of the vertical mixing". But they continue saying that "We have good evidence that this component is relatively small ...". However, in the region centered on the Cape Blanc, which is the region that exports an important amount of nitrogen from the coast to the open ocean, Auger et al. (2016; NSB and SSB regions) show that vertical diffusion is the second most important nitrogen source and sink terms. This term represents about 20% of the inputs in the 0-100 m layer, with sedimentation representing about 50%. The Lagrangian approach*

*thus also contains flaws and uncertainties (see also the list of caveats) and I am not sure that this quantification of the upwelling region's contribution to new production in the NATR and NASE regions adds much more than what the circulation patterns and processes affecting the different biogeochemical provinces tell us.*

We agree with the reviewer that the lack of subgrid vertical mixing is an important caveat that we fully acknowledge in the discussion section (please see line 34, page 25 to line 21, page 26). However, we believe the bias associated with this limitation to likely be small. In fact, given the high spatial and temporal resolution of our velocity field, there is effective vertical mixing occurring as a result of vertical advection and resolved mesoscale and submesoscale dynamics (e.g., secondary ageostrophic circulation around fronts) that is captured by the Lagrangian experiment. Additionally, we have good evidence that this component is likely small. We cite Frischknecht et al. (2018), whose experiment on the California Current System found highly consistent budgets between Eulerian and Lagrangian frameworks. We have added a description on this on page 26, lines 10-15. Nevertheless, we have included an additional statement in the introduction to further stress the disadvantages of the Lagrangian approach (please see lines 30-33, page 3).

*Nevertheless, the authors have shed light on all the questions asked, have taken into account most of the suggestions and have improved the first version of the paper.*

We thank the reviewer for his/her positive comment.

*Some expressions and typos will probably have to be checked during the editing phase. Among them:*

*L13, page 5: "50is" should be "50 m is"*

Done.

*L14, page 7: Fig. 1c and not 1b*

Done.

*L9, page 9: Fig. 1c and not 1b*

Done.